METHODS AND RESOURCES

# Vascularized human cortical organoids (vOrganoids) model cortical development in vivo

Yingchao Shi[1], Le Sun[1], Mengdi Wang[1,2], Jianwei Liu[1], Suijuan Zhong[2], Rui Li[1], Peng Li[1], Lijie Guo[1,3], Ai Fang[1], Ruiguo Chen[1,3], Woo-Ping Ge[4], Qian Wu[2,5]*, Xiaoqun Wang[1,3,6,7]*

**1** State Key Laboratory of Brain and Cognitive Science, CAS Center for Excellence in Brain Science and Intelligence Technology, Institute of Brain-Intelligence Technology (Shanghai), Institute of Biophysics, Chinese Academy of Sciences, Beijing, China, **2** State Key Laboratory of Cognitive Neuroscience and Learning, Beijing Normal University, Beijing, China, **3** University of Chinese Academy of Sciences, Beijing, China, **4** Children's Research Institute, University of Texas Southwestern Medical Center, Dallas, Texas, United States of America, **5** IDG/McGovern Institute for Brain Research, Beijing Normal University, Beijing, China, **6** Institute for Stem Cell and Regeneration, Chinese Academy of Sciences, Beijing, China, **7** Advanced Innovation Center for Human Brain Protection, Beijing Institute for Brain Disorders, Capital Medical University, Beijing, China

☯ These authors contributed equally to this work.
* qianwu@bnu.edu.cn (QW); xiaoqunwang@ibp.ac.cn (XW)

**Data Availability Statement:** All relevant data are within the paper and its Supporting Information files.

**Funding:** This work was supported by the Strategic Priority Research Program of the Chinese

## Abstract

Modeling the processes of neuronal progenitor proliferation and differentiation to produce mature cortical neuron subtypes is essential for the study of human brain development and the search for potential cell therapies. We demonstrated a novel paradigm for the generation of vascularized organoids (vOrganoids) consisting of typical human cortical cell types and a vascular structure for over 200 days as a vascularized and functional brain organoid model. The observation of spontaneous excitatory postsynaptic currents (sEPSCs), spontaneous inhibitory postsynaptic currents (sIPSCs), and bidirectional electrical transmission indicated the presence of chemical and electrical synapses in vOrganoids. More importantly, single-cell RNA-sequencing analysis illustrated that vOrganoids exhibited robust neurogenesis and that cells of vOrganoids differentially expressed genes (DEGs) related to blood vessel morphogenesis. The transplantation of vOrganoids into the mouse S1 cortex resulted in the construction of functional human-mouse blood vessels in the grafts that promoted cell survival in the grafts. This vOrganoid culture method could not only serve as a model to study human cortical development and explore brain disease pathology but also provide potential prospects for new cell therapies for nervous system disorders and injury.

## Introduction

In contrast to the rodent lissencephalic cortex, the human neocortex has evolved into a highly folded gyrencephalic cortex with enormous expansion of the cortical surface and increases in

Academy of Sciences (XDA16020601, XDB32010100) to XW, National Basic Research Program of China (2019YFA0110101 and 2017YFA0103303 to XW; 2017YFA0102601 to QW), the National Natural Science Foundation of China (31671072 to QW; 31771140 and 81891001 to XW), and the Grants of Beijing Brain Initiative of Beijing Municipal Science & Technology Commission (Z181100001518004) to XW. The funders had no role in study design, data collection and analysis, decision to publish, or preparation of the manuscript.

**Competing interests:** The authors have declared that no competing interests exist.

**Abbreviations:** ACSF, artificial cerebral spinal fluid; AIF1, allograft inflammatory factor 1; AMPA, α-amino-3-hydroxy-5-methyl-4-isoxazolepropionic acid; Ang1, angiopoietin-1; APV, DL-2-Amino-5-phosphonopentanoic acid; BBB, blood–brain barrier; BF, bright field; BMI, bicuculline methiodide; BRN2, POU class 3 homeobox 2; CMV, cytomegalovirus; CNQX, 6-Cyano-7-nitroquinoxaline-2,3-dione; CNS, central nervous system; CP, cortical plate; CR, calretinin; CSF, cerebrospinal fluid; CTIP2, chicken ovalbumin upstream promoter transcription factor (COUP-TF)–interacting protein 2; DEG, differentially expressed gene; DGC, dodt gradient contrast; DLX1, distal-less homeobox 1; dpi, days postimplantation; EB, embryonic body; EC, endothelial cell; EOMES, eomesodermin; ExN, excitatory neuron; GABAA, gamma-aminobutyric acid A; GAD1, glutamate decarboxylase 1; GFAP, glial fibrillary acidic protein; GFP, green fluorescent protein; GO, Gene Ontology; GW, gestational week; HBMEC, human brain microvascular endothelial cell; hESC, human embryonic stem cell; HIF1α, hypoxia inducible factor 1 subunit alpha; hiPSC, human-induced pluripotent stem cell; HOPX, homeodomain only protein X; HUN, human nuclear; HUVEC, human umbilical vein endothelial cell; IB4, isolectin I-B4; ImN, immature neuron; IPC, intermediate progenitor cell; iPSC, induced pluripotent stem cell; KCC2, potassium-chloride transporter member 5; KSR, Knockout Serum Replacement; LHX6, LIM homeobox 6; MAP2, microtubule associated protein 2; MBP, myelin basic protein; MGE, medial ganglionic eminence; MGE-div, MGE dividing cell; Mural, mural cells; NEAA, nonessential amino acids; NeuN, RNA binding fox-1 homolog 3; NEUROD2, neuronal differentiation 2; NKX2-1, NK2 homeobox 1; NMDA, N-methyl-D-aspartate; NOD-SCID, nonobese diabetic severe combined immunodeficient; NSC, neural stem cell; OPC, oligodendrocyte progenitor cell; oRG, outer radial

cell type and number [1,2]. Animal models, particularly rodents, have provided significant insight into brain development, but the complexity of the human neocortex cannot be fully captured with these models. Therefore, understanding the genetic changes as well as the mechanistic steps that underpin the evolutionary changes that occur during the development of the neocortex in primates may require new model systems.

Organoids have recently been used to study the development of and pathological changes in different tissue types, such as pancreas, liver, kidney, and retina tissues [3–8]. In addition, several different methods involving the differentiation of human-induced pluripotent stem cells (hiPSCs) have been developed to generate organoids that mimic nervous system development [9–19]. Three-dimensional brain organoids are comprised of multiple cell types that collectively exhibit cortical laminar organization, cellular compartmentalization, and organ-like functions. Therefore, compared to conventional 2D culture, organoids are advantageous because they can recapitulate embryonic and tissue development in vitro and are better at mirroring the functionality, architecture, and geometric features of tissues in vivo.

Previous studies have successfully established suitable approaches for generating cerebral organoids from human embryonic stem cells (hESCs) or hiPSCs that can recapitulate in vivo human cortical development and a well-polarized ventricle neuroepithelial structure that consists of ventricular radial glia (vRGs), outer radial glia (oRGs), and intermediate progenitor cells (IPCs) and the production of mature neurons within layers [12,13,15,17,18,20]. However, a major limitation of current culture approaches that prevents truly in vivo–like functionality is the lack of a microenvironment, such as vascular circulation. Previous studies have reported that the development of the nervous system and the vascular system in the brain is synchronous [21–23]. Vascularization is specifically required for oxygen, nutrient, and waste exchange and for signal transmission in the brain. Additionally, blood vessels around neural stem cells (NSCs) serve as a microenvironment that maintains homeostasis, and they play essential roles in NSC self-renewal and differentiation during embryonic development [24,25]. A lack of vascular circulation can induce hypoxia during organoid culture and accelerate necrosis, which consequently hinders the normal development of neurons and their potential migration [26]. To overcome these limitations, some studies have tried to generate vascularized organoids (vOrganoids) by coculturing hESC- or hiPSC-derived cerebral organoids with endothelial cells (ECs) differentiated from induced pluripotent stem cells (iPSCs) from the same patient [27]. In addition, recent studies have established a robust method for generating vascularized human cortical organoids by introducing hESCs that ectopically express ETV2 into organoids [28]. Mansour and colleagues showed that transplanting human cerebral organoids into the adult mouse brain can result in the formation of a vascularized and functional brain organoid model in vivo [29]. In addition, other reports have demonstrated that compared to transplanting dissociating neural progenitor cells, engrafting cerebral organoids into the lesioned mouse cortex induces enhanced survival and robust vascularization [30]. All of these studies indicate that vascularization is one of the feasible methods to improve organoid survival. In addition to the methods reported in these studies, other stable and reproducible methods are required for establishing vascularized cerebral organoids to model human brain development in vitro and to perform in vivo transplantation.

Here, we developed a 3D culture protocol to generate vOrganoids by coculturing hESCs or hiPSCs with human umbilical vein endothelial cells (HUVECs) in vitro. In our studies, HUVECs were connected and formed a well-developed mesh-like or tube-like vascular system in the cerebral organoids. vOrganoids recapitulated neocortical development, exhibiting different cell types and a neural circuit network, in vitro. In addition, single-cell RNA sequencing (scRNA-seq) analysis verified that vOrganoids shared similar molecular properties and cell types with the human fetal telencephalon. Finally, we intracerebrally implanted vOrganoids

glia; P-gp, P-glycoprotein; PAGA, partition-based graph abstraction; PAX6, paired box 6; PC, principal component; PCA, principal component analysis; PSD95, postsynaptic density protein-95; RELN, reelin; RG, radial glia; SATB2, SATB homeobox 2; scRNA-seq, single-cell RNA sequencing; sEPSC, spontaneous excitatory postsynaptic current; sIPSC, spontaneous inhibitory postsynaptic current; SOX2, SRY-box transcription factor 2; SST, somatostatin; SYB2, synaptobrevin 2; TBR2, eomesodermin; TTX, tetrodotoxin; UMAP, Uniform Mainfold Approximation and Projection; VEGF, vascular endothelial growth factor; vOrganoid, vascularized organoid; vRG, ventricular radial glia; VZ/SVZ, ventricular zone/subventricular zone.

into mice and observed that the grafted vOrganoids survived and integrated into the host cortical tissue in vivo. Importantly, the vessels in vOrganoid grafts connected well with the native blood vessels in the rodents to build a new functional vascularization system. This vOrganoid culture system serves as a model for studying human cortical development and provides new potential therapeutic strategies for treating brain disorders or injuries.

## Results

### Vascularization in the 3D vOrganoid culture system

HUVECs, which are derived from the endothelium of veins from the umbilical cord, have been widely used to explore the function and pathology of ECs [31–33]. In addition, via coculturing with other cell types, HUVECs have been extensively used to characterize angiogenesis during tumorigenesis and other biological processes [34–36]. Due to the tube formation ability of HUVECs (S1A Fig), we generated vascularized cerebral organoids by coculturing hESCs or hiPSCs with HUVECs. Approximately $3 \times 10^6$ dissociated hESCs or hiPSCs and $3 \times 10^5$ HUVECs were plated onto low cell-adhesion plates, and uniformly sized tight embryoid body-like aggregates formed within the first 7 days. On day 18, the aggregates were transferred to petri dishes for neural induction culture. The resulting 3D aggregates were then replated for neural differentiation on day 35. Nonvascularized control organoids were generated by the same workflow except that no HUVECs were added (Fig 1A). vOrganoids differentiated and matured for up to approximately 200 days under the optimized culture conditions (Fig 1B, 1G and 1L).

To visualize vascularization in vOrganoids, LAMININ, the major basement membrane glycoprotein of blood vessels, and isolectin I-B4 (IB4), a marker of ECs, were both used. We observed that the HUVECs connected to form a mesh-like and tube-like structure in vOrganoids as early as 42 days (Fig 1C and S1B Fig, S1 Movie and S2 Movie). Previous studies have confirmed that the proliferating cells in the ventricular zone/subventricular zone (VZ/SVZ) are, on average, closer to blood vessels [37,38]. Interestingly, in our studies, vascularization in vOrganoids primarily occurred just above the VZ/SVZ region, which is enriched with SRY-box transcription factor 2 (SOX2)+ radial glia cells (RGs), homeodomain only protein X (HOPX)+ oRGs, and eomesodermin (TBR2)+ IPCs (Fig 1D and 1E). Meanwhile, some vessels were also detected in the migrating zone and cortical plate (CP; Fig 1D). This localization and distribution were similar to those of blood vessels in the developing human cerebral cortex at gestational week (GW) 12 (S1C Fig). As vOrganoids developed up to day 65, the vascular structures extended into newborn neurons (Fig 1F). The vascular system, mainly located in the outer layers of vOrganoids, which were enriched with neurons, was sustained for over 200 days (Fig 1G).

To test whether the ability of the culture protocol to produce vascular systems in organoids is reproducible in different cell lines, vOrganoids were cultured from two hESC cell lines (H9 and H3) and two hiPSC cell lines (hiPSCs-AE and hiPSCs-LMZ). vOrganoids derived from the different cell lines exhibited similar vascular structures in similar locations (S1D and S1E Fig), indicating the reproducibility of the culture protocol. The vascularization success rate in H9 cells was 100% over more than 30 experiments, while the success rates in the other three cell lines were higher than 95% in at least 3 biological replicates for each. Then, we performed cleaved CASPASE 3 immunostaining in nonvascularized organoids and vOrganoids and found that the cell ratio of cleaved CASPASE 3+ cells in vOrganoids was significantly reduced compared to that in nonvascularized organoids (S1F Fig, upper panel; and S1G Fig). In addition, we also performed immunostaining for hypoxia induciable factor 1 subunit alpha (HIF1α), which is a hypoxia marker. Consistent with the results of cleaved CASPASE 3

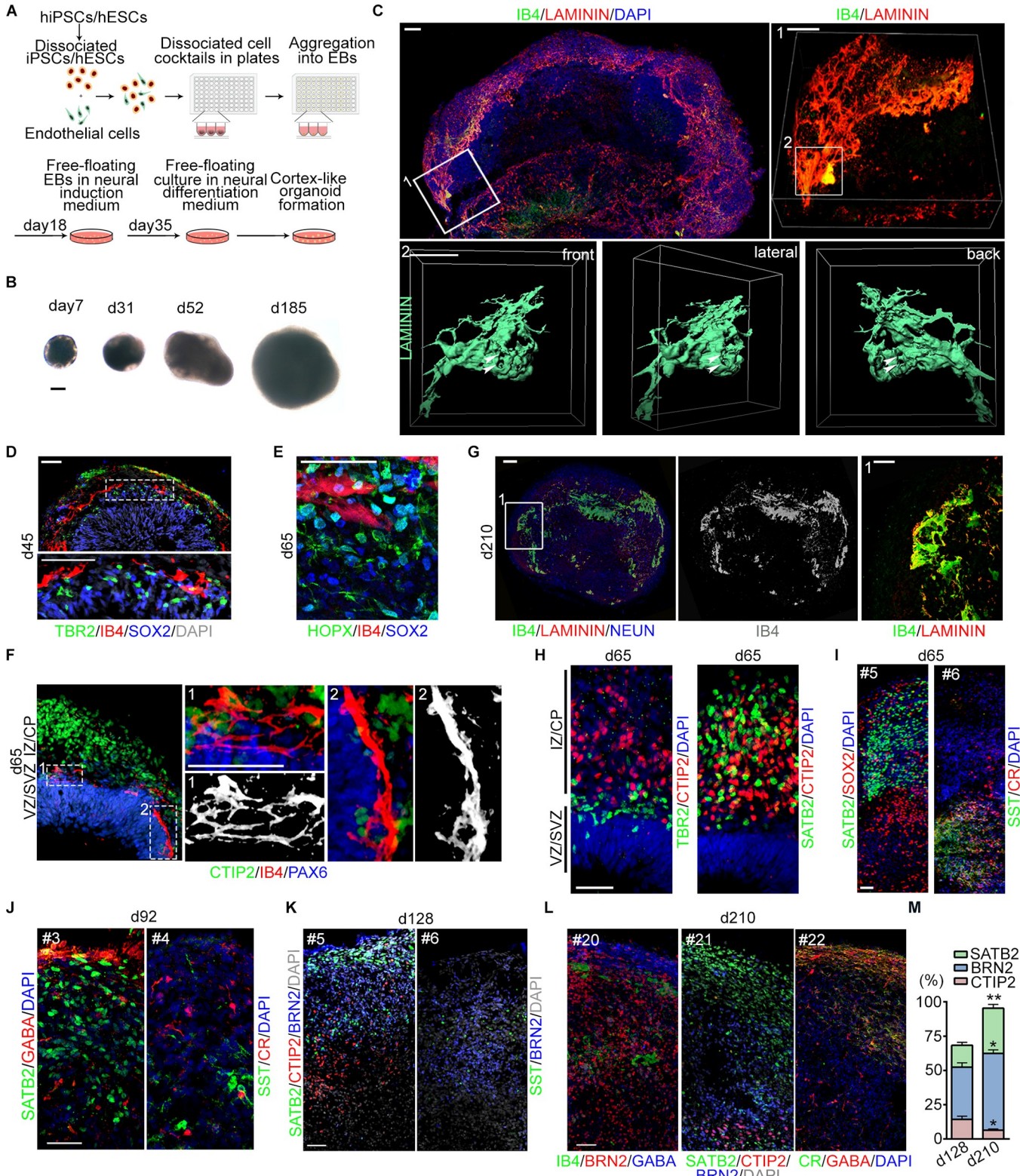

**Fig 1. Cerebral vOrganoids with vascular system recapitulate the cortical spatial organization.** (A) Schematic diagram of the 3D culture methods for generating cerebral organoids with complicate vascular systems. (B) Representative bright field (BF) images of vOrganoids at different stages. Scale bar, 200 μm. (C) Whole mount imaging of vOrganoid on day 42. The elaborate mesh-like vascular systems in vOrganoids were displayed by immunofluorescence staining for LAMININ and IB4. Areas 1 and 2 outlined in boxes were magnified and reconstructed in 3D to depict the complexity of vasculature in vOrganoids. Scale

bar, 100 μm (upper left), 50 μm (in box 1), 50 μm (in box 2). The arrowheads pointed out the hollows in the vascular systems that are permeable at different views. (D) Representative immunofluorescence staining figure for TBR2, SOX2, and IB4 to reveal that the vasculogenesis in vOrganoids is synchronous to the neurogenesis at early stage. Scale bar, 50 μm. (E) Representative immunofluorescence staining figure for HOPX, SOX2, and IB4 to demonstrate that the HOPX$^+$ SOX2$^+$oRG cells could be detected in the vOrganoids at day 65. Scale bar, 50 μm. (F) Representative immunofluorescence staining figure for CTIP2/ IB4/PAX6 at day 65 to demonstrate that the IB4$^+$ vascular structures would progressively extend into newborn neurons (CTIP2$^+$) with the development of vOrganoids. Scale bar, 50 μm. (G) Representative IB4 and LAMININ staining figure in vOrganoid at day 210 to demonstrate that the vascular system could be maintained for over 200 days. Scale bars, 100 μm. (H) The spatial organization of vOrganoids was illustrated by immunofluorescence staining for TBR2/CTIP2 (left panel) and SATB2/CTIP2 (right panel) at day 65. Scale bar, 50 μm. (I) Representative immunofluorescence staining figure for the SATB2 and SOX2 to illustrate that SATB2$^+$ cells are mainly located above the SOX2$^+$ progenitor cells (left panel); SST and CR staining illustrated the emergence of interneurons in vOrganoids at day 65 (right panel). The "#5" and "#6" labelled in the upper left represent the number of continuous sections of vOrganoids. Scale bars, 50 μm. (J-L) Representative immunostaining figure for the pyramidal layer markers and interneuron markers in the continuous cryosections of vOrganoids at day 92 (J), day 128 (K), and day 210 (L). Scale bars, 50 μm. (M) The percentage of SATB2$^+$, BRN2$^+$, and CTIP2$^+$ cells in the vOrganoids of day 128 and day 210, respectively. $n$ = 3 organoids from three independent experiments. All data are presented as means ± SEM, independent-samples $t$ test, $^*p < 0.05$, $^{**}p < 0.01$. The numerical data underlying this figure can be found in the Fig 1M sheet of S1 Data. BRN2, POU class 3 homeobox 2; CR, calretinin; CTIP2, chicken ovalbumin upstream promoter transcription factor (COUP-TF)–interacting protein 2; EB, embryonic body; hESC, human embryonic stem cell; hiPSC, human-induced pluripotent stem cell; HOPX, homeodomain only protein X; IB4, isolectin I-B4; iPSC, induced pluripotent stem cells; oRG, outer radial glia; PAX6, paired box 6; SATB2, SATB homeobox 2; SOX2, SRY-box transcription factor 2; SST, somatostatin; TBR2, eomesodermin; vOrganoid, vascularized organoid.

staining, the cell ratio of HIF1α-positive cells in the vOrganoids was significantly reduced (S1F Fig, lower panel, and S1G Fig). The HIF1α-positive cells in vOrganoids were mainly located in the center of the organoids, while those in nonvascularized organoids were abundant and widespread (S1F Fig, lower panel). Moreover, vOrganoids were larger in size and possessed thicker neuroepithelia than nonvascularized organoids (S1H–S1K Fig). These results indicate that the vascular systems in vOrganoids may provide more oxygen to support cell survival, resulting in decreased cell death and larger vOrganoids. Human brain microvascular endothelial cells (HBMECs) are the main ECs in the human brain and play important roles in the development of the blood–brain barrier (BBB) [39]. Given the difference between HUVECs and HBMECs, we also examined whether coculture can induce HUVECs to adopt a more brain-like EC fate. P-glycoprotein (P-gp), an efflux transporter that influences the absorption, distribution, and elimination of a variety of compounds, has been reported to be expressed on brain capillary ECs. In our studies, P-gp was abundantly detected in brain capillary ECs at GW12 but not in cultured HUVECs (S1L Fig and S1M Fig). These results indicate that when cultured individually in the absence of organoids, HUVECs can form tube-like structures that are similar to those in vOrganoids, but do not express P-gp (S1M Fig). Next, we examined the expression of P-gp in the vOrganoids on day 83. The high degree of IB4 and P-gp colocalization in vOrganoids (S1N Fig) indicated that coculture might induce HUVECs to adopt a more brain-like EC fate.

## Recapture of cell subtypes during neurogenesis

To determine whether vOrganoids can recapitulate human cortical organization (S1O Fig), we stained for cortical layer markers on day 65. We found that TBR2$^+$ IPCs were adjacent to the early CP, which was indicated by chicken ovalbumin upstream promoter transcription factor (COUP-TF)–interacting protein 2 (CTIP2)$^+$ cells (Fig 1H, left panel). Furthermore, we found that some SATB homeobox 2 (SATB2)$^+$ cells were superficially localized to CTIP2$^+$ neurons and some were widely distributed in the migrating zone and newly formed CPs in vOrganoids (Fig 1H, right panel; Fig 1I, left panel; and S1P Fig). This observation was consistent with reports showing that SATB2 is expressed by virtually all upper layer projection neurons as well as 30% of deep layer neurons that project through the corpus callosum [40–42]. In addition, we also observed that early-born reelin (RELN)$^+$ cells were located in the superficial layer while later-born TBR1$^+$ cells were located in the deeper layer (S1Q Fig).

Because pyramidal neurons and interneurons are both required to form neural circuits, we also paid close attention to the different subtypes of interneurons in vOrganoids. Calretinin (CR)$^{+}$ or somatostatin (SST)$^{+}$ cells were first detectable at day 65 in continuous sections of vOrganoids (Fig 1I, right panel). Moreover, layer-specific pyramidal neurons were surrounded by sparse interneurons in the late stages of the 3D culture on days 92, 128, and 210 (Fig 1J–1L). The proportion of upper layer neurons increased from 15.94% to 33.04% from day 128 to day 210, and the interneurons were sparsely distributed (Fig 1K–1M), indicating that the vOrganoid culture system models the organization of the neocortex in vivo.

Next, we used scRNA-seq (10X Genomics Chromium) to capture the cellular and molecular features of the vOrganoids. We collected nonvascularized organoid and vOrganoid samples at two time points: day 65 and day 100 (Fig 2A and 2B and S2A Fig). There were three independent experimental replicates for each group (S2A Fig). After quality control, we obtained a final dataset of 57,180 cells from 12 independent samples encompassing an average of 2,762 genes per cell (S2B Fig, S1 Table). To characterize the cellular heterogeneity of vOrganoids and nonvascularized organoids, we performed Uniform Mainfold Approximation and Projection (UMAP) analysis with Seurat (Stuart and colleagues, 2019) and identified 11 major clusters, including RGs, oRGs, IPCs, cell cycle active cells (labeled as cell cycle), immature neurons, excitatory neurons, interneurons, microglia, astrocytes, oligodendrocyte progenitor cells (labeled as OPCs) and choroid plexus cells, based on the expression of classical gene markers (Fig 2A, S2C and S2D Fig, and S1 Table). The high transcriptomic correlations between the corresponding cell types in the organoids suggested that there are no remarkable cell type differences between nonvascularized organoids and vOrganoids (S2E Fig).

We then aimed to compare the similarities between the organoids and the human fetal telencephalon at the single-cell level [43]. To identify the homologous cell types between organoids and primary tissue, we integrated organoid cells and embryonic human telencephalon cells and found that the same cell types were generally close to each other (Fig 2C). To clearly exhibit the correlations of cell types between organoids and human telencephalon, we constructed a correlation network via partition-based graph abstraction (PAGA) (Fig 2D and 2E and S2F Fig). In the PAGA plots, the RGs, oRGs, IPCs, excitatory neurons, interneurons, astrocytes, microglia, choroid plexus cells, and OPCs in the organoids were located close to the corresponding cell types in the human telencephalon (Fig 2D and 2E and S2F Fig). These results suggest that based on transcriptome analysis, the cell types in organoids are highly similar to the corresponding cell types in the human fetal telencephalon.

To explore whether vOrganoids more closely recapitulate fetal telencephalon development than control organoids, we compared the single-cell transcriptome profile of the human fetal telencephalon with that of control organoids (S2G and S2H Fig) and vOrganoids (S2I and S2J Fig) separately. Based on PAGA as well as transcriptomic correlation analysis, the cell types in control organoids as well as those in vOrganoids were highly similar to those in the human fetal telencephalon (S2K–S2N Fig). Although there were no obvious differences in cell types or the similarity to the human fetal telencephalon between vOrganoids and control organoids, we then compared the development of vOrganoids and nonvascularized organoids by analyzing the scRNA-seq data. First, we predicted the developmental states of the cells in control organoids and vOrganoids along the pseudo-time trajectory by monocle analysis [44–46]. The fitted curves of the cell density along the pseudo-time were plotted on day 65 and day 100 (Fig 2F). More vOrganoid cells than nonvascularized organoid cells were assigned a high pseudo-time score (Fig 2F). The extent of divergence was more obvious on day 65 than on day 100 (Fig 2F). These results indicate that vOrganoids develop faster than nonvascularized organoids, even though this difference decreases during development. To further validate this observation, we calculated the maturation score of each cell in vOrganoids and controls (Fig

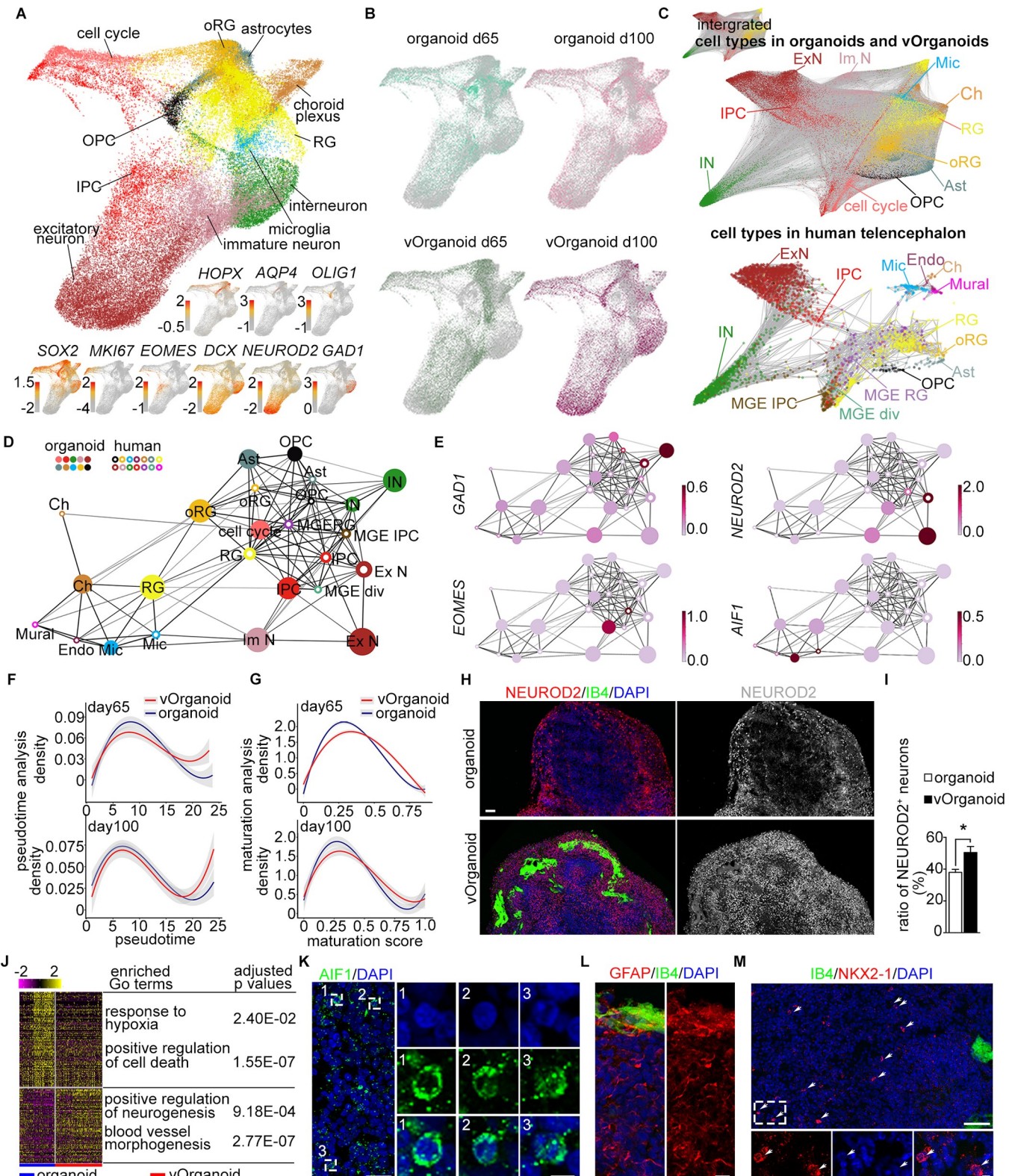

**Fig 2. Cell type mapping between the scRNA-seq data of organoids and human fetal telencephalon.** (A) Visualization of the major cell types in organoids and vOrganoids by 3D UMAP. Each dot represents one individual cell and colored by cell types. The expression of known gene markers was visualized at the lower panel and cells were colored by the expression level (red, high; gray, low). (B) The cell distributions of control organoids and vOrganoids at d65 and d100, respectively, were individually showed using UMAP. Each dot represents one individual cell and colored by sample information. (C) The scRNA-seq data

of organoids and human fetal telencephalon were integrated to display the cell similarities in forced-directed graph. The integrated data were showed in the top left zoomed-out plot. Meanwhile, the cell distributions of organoids and human fetal telencephalon were highlighted separately. Each dot represents a single cell and is colored according to the cell types in the integrated data. The width of edges is scaled with cell–cell connectivity. (D) The correlations among cell types are displayed in the PAGA graph. The cell types in the organoids and human fetal cortex datasets were denoted by the solid and hollow dots, respectively. And the same cell types in two datasets were colored identically. The size of dots was scaled with the cell numbers; width of edges was scaled with the connectivity between cell types. (E) The expression of the well-known gene markers in different cell types is showed in the PAGA plots. The GAD1, NEUROD2, EOMES, and AIF1 are the specific genes for interneuron, excitatory neurons, IPCs, and microglia, respectively. Nodes are colored according to the gene expression levels (light pink, low; dark red, high). (F) The fitted curves of cell density along the pseudo-time at day 65 (upper) and day 100 (lower) The shadow represents the confidence interval around the fitted curve. (G) The curves of cell density along the pseudo-maturation trajectory at day 65 (upper) and day 100 (lower). The shadow represents the confidence interval around the fitted curve. (H) Immunofluorescence staining for NEUROD2/IB4 in the control organoids and vOrganoids at day 65. The expressions of NEUROD2 are individually displayed in the right panel. Scale bar, 100 μm. (I) The ratio of NEUROD2$^+$ excitatory neurons is higher in the vOrganoids than in the control organoids at day 65. $n$ = 4, 4 control organoids and vOrganoids from three independent experiments. (J) Heatmap showing the expression of DEGs between the progenitor cells (included RG, oRG, cell cycle active cells, and IPCs) of control organoids and vOrganoids at day 65. The enriched gene ontology of the DEGs and the adjusted $p$-values are also listed. (K) Representative immunofluorescence staining figure for AIF1, a specific microglia marker, to illustrate the presence of microglia in organoids. Scale bar, 10 μm (left), 5 μm (right). (L) Immunofluorescence staining for GFAP to illustrate the presence of astrocytes in organoids. Scale bar, 10 μm. (M) A few of NKX2-1–positive cells are detected in the immunofluorescence staining section. The area in the white dashed box is magnified in the lower panel. Scale bar, 200 μm (upper), 20 μm (lower). The numerical as well as metadata underlying this figure can be found in the Fig 2A, 2B, 2C and 2I sheets of S1 Data. AIF1, allograft inflammatory factor 1; Ast, astrocyte; cell cycle, cell cycle active cell; Ch, choroid plexus; DEG, differentially expressed gene; Endo, endothelial cell; EOMES, eomesodermin; ExN, excitatory neuron; GAD1, glutamate decarboxylase 1; GFAP, glial fibrillary acidic protein; IB4, isolectin I-B4; ImN, immature neuron; IN, interneuron; IPC, intermediate progenitor cell; MGE, medial ganglionic eminence; MGE-div, MGE dividing cell; Mic, microglia; Mural, mural cell; NEUROD2, neuronal differentiation 2; NKX2-1, NK2 homeobox 1; OPC, oligodendrocytes progenitor cell; oRG, outer radial glia; PAGA, partition-based graph abstraction; RG, radial glia cell; scRNA-seq, single-cell RNA sequencing; UMAP, Uniform Mainfold Approximation and Projection; vOrganoid, vascularized organoid.

2G). Similar to the developmental pseudo-time analysis, there were fewer vOrganoid cells than nonvascularized organoid cells with low maturation scores and more vOrganoid cells than nonvascularized organoid cells with high scores (Fig 2G). Together, both the developmental state and maturation trajectory analyses suggest that the vascular systems in vOrganoids might accelerate the development of vOrganoids in the early stage.

Next, to understand why vOrganoids develop faster than control organoids, we analyzed the cell composition on day 65. We found that the percentage of one type of neural progenitor cell—oRGs—was much higher in vOrganoids than in nonvascularized organoids (S2O Fig). oRGs originate from RGs and play important roles in neurogenesis, greatly contributing to rapid excitatory neuron production during human fetal cortical development [47,48]. Accordingly, the percentage of excitatory neurons, the progenies of neural progenitor cells, in vOrganoids was much higher than that in nonvascularized organoids on day 65 (S2O Fig). Consistently, more neuronal differentiation 2 (NEUROD2)+ neurons were detected in vOrganoids than nonvascularized organoids by immunostaining on day 65 (Fig 2H and 2I). Thus, both scRNA-seq and immunostaining suggested that the vascular systems in vOrganoids may promote neurogenesis in vitro.

Given the intimate interactions between the vasculature and proliferative progenitor cells during brain development [49–52], we also attempted to detect whether the vascular systems in vOrganoids can influence gene expression in neural progenitors. We conducted the differentially expressed gene (DEG) and Gene Ontology (GO) enrichment analysis of progenitors (cell cycle active cells, RGs, oRGs, and IPCs) in vOrganoids and nonvascularized organoids. The GO terms enriched in progenitor cells in nonvascularized organoids were responses to hypoxia and positive regulation of cell death (Fig 2J, S1 Table). These GO terms are consistent with the higher number of HIF1α$^+$ and cleaved CASPASE 3$^+$ cells observed in nonvascularized organoids (S1F and S1G Fig). Meanwhile, the GO terms enriched in progenitor cells in vOrganoids were positive regulation of neurogenesis and blood vessel morphogenesis (Fig 2J, S1 Table).

In addition to neurons, we also detected glial cells, such as microglia, OPCs, and astrocytes, in our scRNA-seq dataset (Fig 2A). We performed immunostaining for microglia-specific allograft inflammatory factor 1 (AIF1) and astrocyte-specific glial fibrillary acidic protein (GFAP)

and verified the presence of these markers in vOrganoids (Fig 2K and 2L). In addition, we detected interneurons with high expression of glutamate decarboxylase 1 (GAD1)/distal-less homeobox 1 (DLX1) and sparse expression of NK2 homeobox 1 (NKX2-1) and LIM homeobox 6 (LHX6) in our scRNA-seq data (Fig 2A and S2C Fig). The immunostaining results also illustrated sparsely distributed NKX2-1+ cells in vOrganoids (Fig 2M). Notably, we also detected a cluster of choroid plexus epithelial cells, which displayed transcriptomic similarity with the choroid plexus in the human telencephalon (Fig 2A, S2C Fig and S2F Fig). In vivo, the choroid plexus is mainly responsible for the production of cerebrospinal fluid (CSF), which serves as a rich source of proteins, lipids, hormones, cholesterol, glucose, microRNA, and many other molecules for the maintenance of central nervous system (CNS) functions and plays a role in embryonic neurogenesis [53].

## Modeling functionality maturation during neurogenesis

Several previous studies have illustrated that neurons in cortical organoids without vascular structures are ultimately able to reach mature states as their morphology and functionality progressively mature [13,20,54]. Because vOrganoids partially recapitulate human cortical development based on gene expression and cell subtypes, we next investigated the functional characteristics of the neurons by obtaining electrophysiological recordings. We performed patch-clamp recordings on slices from both control organoids and vOrganoids. To unbiasedly record neurons in vOrganoids and nonvascularized organoids, we selected neurons with similar soma diameters (S3A Fig) located within 150 μm of the edge of the organoid slices (S3B Fig). There were no differences in the amplitude of the inward currents. However, dramatic increases in outward current amplitudes were found in vOrganoid cells on day 80 (Fig 3A and 3B and S3C Fig). The increased outward current might in turn contribute to higher firing ability under current stimulation of different amplitudes in vOrganoids compared to nonvascularized organoids (S3D and S3E Fig). Because outward current increases significantly during the maturation process of cortical neurons [55], our observations suggest that neurons in vOrganoids might be more mature than those in nonvascularized organoids. Apart from increased outward current, vOrganoid neurons also exhibited lower resting membrane potential (S3F Fig) and greater cell capacitance (S3G Fig), suggesting that there might be increased dendritic or axonal growth in vOrganoids. Additionally, more spontaneously active cells were detected in vOrganoids (8/56 cells) than in nonvascularized organoids (4/52 cells) on day 80 (S3H Fig). Together, these results indicate that the vascular system in vOrganoids may accelerate the progression of the functional development of individual neurons in vitro.

To build coordinated neural circuits, individual neurons must establish synaptic connections with each other. Therefore, we next focused on the formation of synapses in the vOrganoids. We recorded spontaneous excitatory postsynaptic currents (sEPSCs) and sIPSCs from vOrganoids from day 90 to day 100 (Fig 3C–3F), and sIPSCs were blocked by the gamma-aminobutyric acid A (GABAA) receptor antagonist bicuculline methiodide (BMI) (S3I Fig). Furthermore, to monitor the formation of the synapses, staining for synaptobrevin 2 (SYB2) and postsynaptic density protein-95 (PSD95) was performed to identify presynapses and postsynapses, respectively. We found abundant SYB2+ and PSD95+ puncta, some of which were colocalized, in vOrganoids on day 62 (Fig 3G). On day 210, dense distribution of RNA binding fox-1 homolog 3 (NeuN)+/microtubule associated protein 2 (MAP2)+ mature neurons was observed (S3J Fig), and Na+ currents and sIPSCs/sEPSCs were captured from these mature neurons (S3K–S3M Fig), which was indicative of long-standing synaptic formation in vOrganoids.

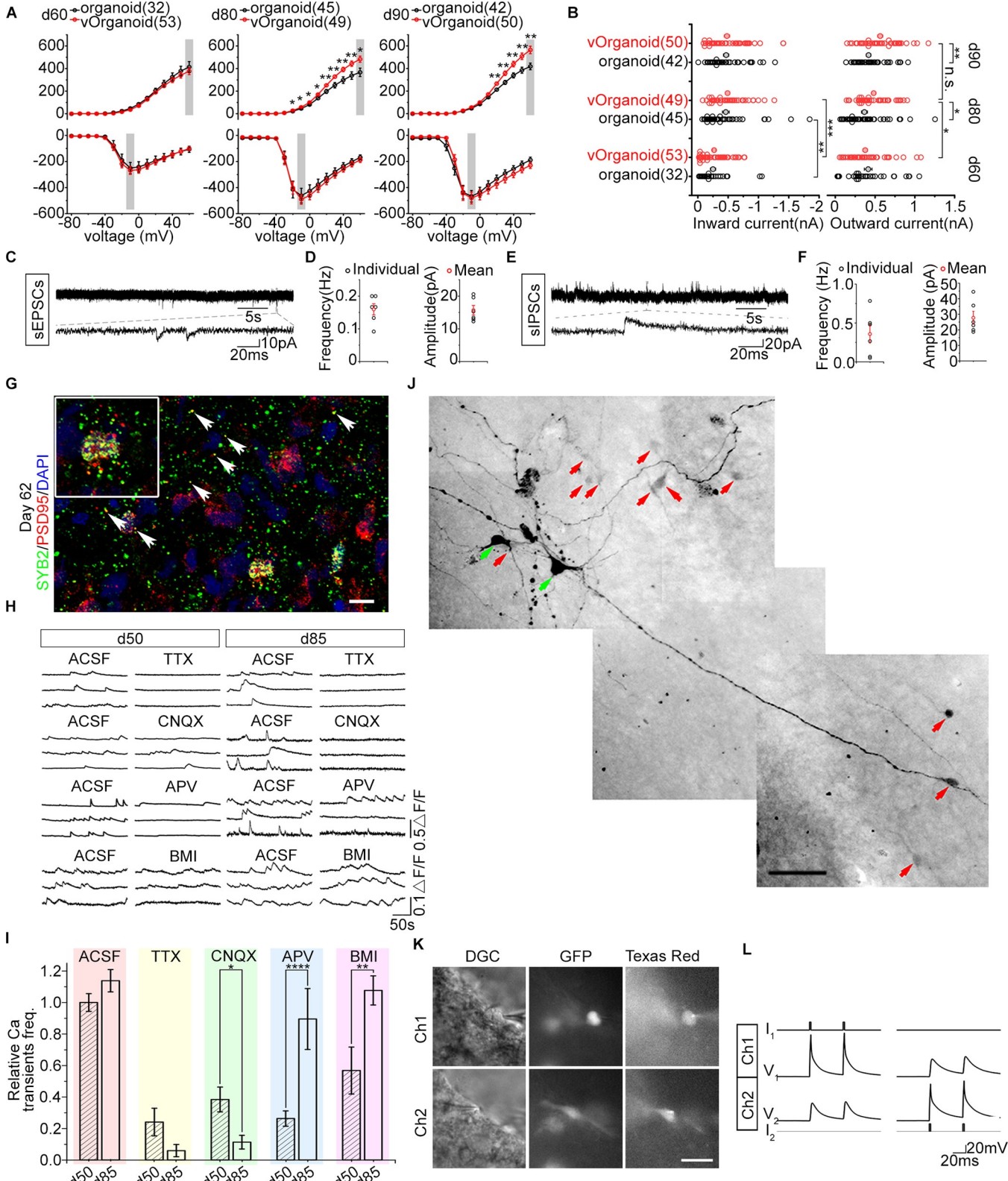

**Fig 3. Electrophysiological properties of cells in the vOrganoids at different developmental stages.** (A) The amplitudes of outward (upper) and inward (lower) currents from the organoids (black) and vOrganoids (red) cells at day 60 (32 cells from 5 organoids in 4 independent experiments; 53 cells from 8 vOrganoids in 6 independent experiments), day 80 (45 cells from 6 organoids in 4 independent experiments; 49 cells from 6 vOrganoids in 4 independent

experiments, *p*-values: 0.0428, 0.0278, 0.0222, 0.0171, 0.0085, 0.0066, 0.0072, 0.0099, and 0.0131 when evoked voltages are −20, −10, 0, 10, 20, 30, 40, 50, and 60 mV) and day 90 (42 cells from 6 organoids in 4 independent experiments; 50 cells from 6 vOrganoids in 5 independent experiments, *p*-values: 0.0090, 0.0027, 0.0033, 0.0025, and 0.0018 when evoked voltages are 20, 30, 40, 50, and 60 mV). Two-sample *t* test, data shown as mean ± SEM, *$^*p < 0.05$, $^{**}p < 0.01$, $^{***}p < 0.001$. (B) The amplitudes of outward currents elicited by +60 mV (indicated by gray box in upper panel of A) and the amplitudes of inward currents elicited by −10 mV (indicated by gray box in lower panel of A) from organoid (black) and vOrganoid (red) cells at day 60, day 80, and day 90. Open circles indicate the amplitude of current from individual cells. Filled circles indicate the mean value. *p*-Values: 0.0108 (outward current of vOrganoid d60 versus d80), 0.0615 (outward current of vOrganoid d80 versus d90), 0.0131 (outward current at day 80 organoid versus vOrganoid), 0.0018 (outward current at day 90 organoid versus vOrganoid), 0.0066 (inward current of organoid d60 versus d80), 0.00001 (inward current of vOrganoid d60 versus d80). Two-sample *t* test, data shown as mean ± SEM, $^*p < 0.05$, $^{**}p < 0.01$, $^{***}p < 0.001$. (C-F) Spontaneous EPSCs (C) and IPSCs (E) were recorded in vOrganoids. The average frequency and amplitude of EPSCs and IPSCs are shown in (D) and (F), respectively. *n* = 6, 6 cells from 3 organoid and 3 vOrganoids in three independent experiments for sEPSCs (D) and sIPSCs (F), respectively. (G) Synapses in vOrganoids are revealed by immunofluorescence staining for the pre- and postsynaptic markers, SYB2 and PSD95, respectively. Scale bar, 10 μm. (H) Intracellular spontaneous calcium fluctuations were measured before and after the application of TTX, CNQX, APV, and BMI in the vOrganoids at day 50 and day 85, respectively. (I) The effects of different treatments on spontaneous calcium fluctuations were compared between day 50 and day 85 in the statistical results. Relative Ca2$^+$ transient frequencies are normalized with the ACSF data of day 50. There are 188 and 124 cells in total for day 50 and day 85. All data are presented as means ± SEM. *n* = 188, 124, 31, 9, 65, 33, 77, 36, 15, and 46 from three independent experiments. Two-sample *t* test, p_ACSF = 0.1266, p_TTX = 0.2798, p_CNQX = 0.0208, p_APV = 0.00005, p_BMI = 0.0075. $^*p < 0.05$, $^{**}p < 0.01$, $^{***}p < 0.001$, $^{****}p < 0.0001$. (J) Coupling pattern was visualized by cell injection. Green arrows indicated the injected cells. Red arrows indicated the cells that were connected to the injected cells. Scale bar, 50 μm. (K-L) The gap junctions between two cells (CMV-GFP labeled) were identified by dual-patch recording. The morphology of cells that dual patched was showed in (K). Scale bar, 50 μm in (K). The voltage deflections with small amplitudes were recorded in one cell while currents were injected into the other cell (L). The numerical data underlying this figure can be found in the Fig 3A, 3B, 3D, 3F and 3I sheets of S1 Data. ACSF, artificial cerebral spinal fluid; APV, DL-2-Amino-5-phosphonopentanoic acid; BMI, bicuculline methiodide; CMV, cytomegalovirus; CNQX, 6-Cyano-7-nitroquinoxaline-2,3-dione; DGC, dodt gradient contrast; GFP, green fluorescent protein; PSD95, postsynaptic density protein-95; sEPSC, spontaneous excitatory postsynaptic current; sIPSC, spontaneous inhibitory postsynaptic current; SYB2, synaptobrevin 2; TTX, tetrodotoxin; vOrganoid, vascularized organoid.

Intracellular calcium fluctuation imaging is an efficient method for characterizing the functional features of neural activities in cortex-like organoids. Therefore, we performed calcium dye imaging to detect calcium oscillations and observed spontaneous calcium surges in individual cells. When we blocked the action potentials with the application of tetrodotoxin (TTX), a specific blocker of voltage-gated sodium channels, we observed dampened calcium surges in most cases, indicating that spontaneous calcium events in neurons depend on neuronal activity (Fig 3H and S3N Fig). To further characterize the types of mature neurons, we applied exogenous glutamate and observed more frequent calcium events, indicating that these neurons exhibited glutamatergic receptor activity on day 85 (S3O Fig). Given the predominance of glutamatergic neurons in the aggregates, we combined a pharmacological assay with calcium imaging to further illustrate the differences in receptor-mediated synaptic transmission on day 50 and day 85. Neuronal activity in vOrganoids was reduced during treatment with DL-2-Amino-5-phosphonopentanoic acid (APV), which is an N-methyl-D-aspartate (NMDA) receptor antagonist, 6-Cyano-7-nitroquinoxaline-2,3-dione (CNQX), which is an α-amino-3-hydroxy-5-methyl-4-isoxazolepropionic acid (AMPA) receptor antagonist, or BMI, which is a GABAA receptor antagonist, indicating that NMDA receptor–, AMPA receptor–, and GABAA receptor–mediated synaptic activities were present after day 50 (Fig 3H and 3I). Importantly, when we compared neurons on day 50 and day 85, we observed reduced sensitivity to APV and greater sensitivity to CNQX on day 85 (Fig 3I), which indicated a transition from NMDA receptor–to AMPA receptor–mediated excitatory synaptic activity during the integration and maturation of vOrganoids. This is similar to what happens in vivo [56]. We also observed that BMI severely reduced neural activity on day 50 but not on day 85 (Fig 3H and 3I), indicating that the GABAergic response changed from depolarizing to hyperpolarizing during development; this change can be induced by a decreased intracellular chloride ion concentration due to the expression of potassium-chloride transporter member 5 (KCC2) [57].

In addition to the formation of chemical synapses, we also investigated the formation of electrical synapses (gap junctions) in vOrganoids. To identify the connectivity of the neural network, we explored how many cells were connected to one neuron through gap junctions.

When we injected neurobiotin into two neurons, 11 coupled neurons were observed (Fig 3J and S3P Fig), suggesting that electrical connections exist in vOrganoids. To directly observe electrical coupling between neurons, we performed dual-patch recordings of vOrganoids. When the dual patch was established, voltage deflections with small amplitudes were recorded from one cell while currents were injected into the other cell (Fig 3K and 3L). Bidirectional electrical transmission indicated the existence of gap junctions between these two cells (Fig 3L). Together, these results indicate that the neurons in vOrganoids can become functionally mature with the emergence of a spontaneous action potential and that functionally mature neurons can later connect to one another based on the formation of abundant chemical and electrical synapses and receptor maturation.

## Transplantation of vOrganoids reconstructs the vascular system in the mouse cortex

The long-term survival of organoid grafts in a host requires vascularization to satisfy the adequate delivery of oxygen and nutrients. The intracerebral implantation of hiPSC-derived brain organoids into the retrosplenial cortex of immunodeficient mice results in the recruitment of mouse blood vessels that grow into the grafts and support the long-term survival of the cells [29]. Therefore, we next tested whether the vascular systems in the vOrganoids can connect to blood vessels in the brain of the host to build a functional circulatory system. Sixty-day-old vOrganoids were intracerebrally implanted into a cavity that was made in the S1 cortex of non-obese diabetic severe combined immunodeficient (NOD-SCID) mice (Fig 4A). At 60 days postimplantation (dpi), we observed the integration of the organoid grafts into the host brain tissue (Fig 4B). SATB2$^+$ cells were generally distributed close to the surface even though sometimes interrupted by SATB2$^-$/CTIP2$^-$ VZ-like rosettes. Meanwhile, CTIP2$^+$ cells were mainly located in the deep area of the grafts (Fig 4B).

To test whether synaptic integration emerges in grafted vOrganoids, we performed immunofluorescence staining for SYB2 and PSD95 to observe presynaptic and postsynaptic proteins, respectively, in the graft area close to the graft–host border at 60 dpi. The colocalization and close association of SYB2 and PSD95 indicated that there was synaptic connectivity in the grafts (Fig 4C). However, the presynaptic and postsynaptic sources could not be distinguished in the current study.

Because vOrganoids showed less cell death than nonvascularized organoids in culture, we next asked whether vOrganoid grafts survive better than nonvascularized grafts. We implanted control organoids or vOrganoids into the S1 cortex of NOD-SCID mice and performed immunostaining for cleaved CASPASE 3 in grafts at 60 dpi. Less cell death was observed in the vOrganoid grafts than nonvascularized organoid grafts (S4A and S4B Fig). Additionally, we also compared cell death in organoid grafts with that in in vitro cultured vOrganoids on the same day (day 120). On day 120, there were fewer cleaved CASPASE 3$^+$ cells in cultured vOrganoids than control nonvascularized grafts but significantly more in cultured vOrganoids than vOrganoid grafts (S4A and S4B Fig). These results suggest that the vascular systems in vOrganoids can improve cell survival in the host brain.

We next injected Alexa Fluor 594–conjugated dextran into the mouse caudal vein to observe the blood flow in organoid grafts by live two-photon microscopy. Interestingly, steady blood flow was captured in organoid grafts, indicating the formation of a functional vascular system between the graft and host (S3 Movie). The 3D reconstructed images also briefly showed the vascular systems in the grafts (Fig 4D, S4 Movie). In addition, to further demonstrate that the ECs in the grafts had reliably integrated into the new vascular system, immunofluorescence staining for dextran and LAMININ was performed to label blood vessels, while

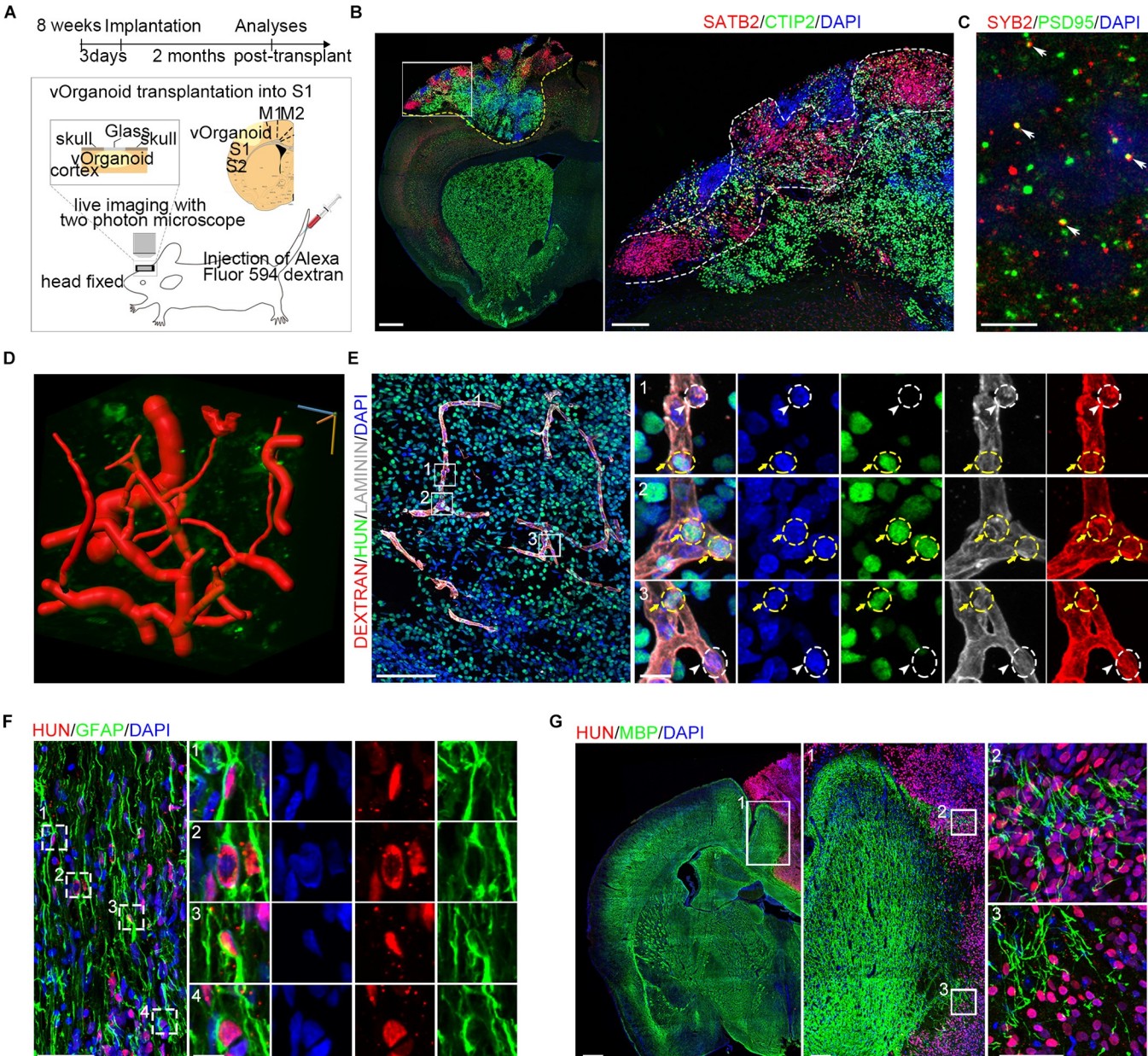

**Fig 4. The vOrganoids play important roles in the reconstruction of vascular system after transplantation. (A)** Schematic diagram to demonstrate the vOrganoids implantation protocols in our studies. The vOrganoids were transplanted into the S1 cortex of NOD-SCID mice. **(B)** Representative immunofluorescence staining figure for the classical cortical layer markers, CTIP2 and SATB2, at 2 months postimplantation. The boxed area was magnified in the right panel. Scale bar, 500 μm (left), 50 μm (right). **(C)** Representative immunofluorescence staining figure for presynaptic (SYB2) and postsynaptic (PSD95) in the vOrganoid grafts of 60 dpi. The displayed area was close to the graft–host border. The colocalization (arrowheads) and close association of SYB2 and PSD95 indicate that synaptic connectivity emerges in the organoid grafts. Scale bar, 10 μm. **(D)** The blood vessels in the vOrganoid grafts were reconstructed in three dimensions. The red tubular structure was blood vessels. Scale bar was labeled at three dimensions. Scale bar, 30 μm. **(E)** Immunofluorescence staining for LAMININ, dextran, and HUN were performed to confirm that the human ECs derived from vOrganoids (labeled by yellow circles and yellow arrows) and the mice ECs derived from hosts (labeled by white circles and white arrowheads) were both detected in the vascular systems in the vOrganoid grafts. Scale bar, 100 μm (left), 10 μm (right). **(F)** Representative immunofluorescence staining figure for GFAP and HUN in the vOrganoid grafts. A few HUN-positive astrocytes were labeled by boxes and magnified in right panels. Scale bar, 50 μm (left), 10 μm (right). **(G)** Representative immunofluorescence staining figure for MBP and HUN to illustrate that myelinization just could be infrequently observed at the border of graft-host commissure. The boxed areas were magnified. Scale bar, 500 μm (left), 100 μm (middle), 50 μm (right). See also S4 Fig. CTIP2, chicken ovalbumin upstream promoter transcription factor (COUP-TF)–interacting protein 2; dpi, days postimplantation; GFAP, glial fibrillary acidic protein; HUN, human nuclear; MBP, myelin basic protein; NOD-SCID, nonobese diabetic severe combined immunodeficient; PSD95, postsynaptic density protein-95; SATB2, SATB homeobox 2; SYB2, synaptobrevin 2; vOrganoid, vascularized organoid.

human nuclear (HUN) was used to distinguish vOrganoid grafts from host cells. We observed that human HUVEC-derived HUN⁺ ECs and mouse HUN⁻ ECs coexisted in the blood vessels in vOrganoid grafts 2 months after implantation (Fig 4E).

A study by Mansour and colleagues demonstrated that blood vessels from the host can grow into organoid grafts to support their long-term survival [29]. We next asked whether angiogenesis appears earlier in the vOrganoid grafts than in nonvascularized organoid grafts. Thus, we performed immunostaining for IB4 and HUN in organoid grafts at 30 dpi, a time point at which the blood vessels from the host did not extensively grow into the organoid grafts (S4C Fig). Some IB4⁺ blood vessels were detectable in the center of the vOrganoid grafts, while the majority of blood vessels in the nonvascularized organoid-implanted brains were distributed close to the graft–host borders (S4C Fig).

Considering the supporting roles of glial cells in normal neural activities and the integration of the grafts into host tissue, we next examined the level of gliogenesis within the organoid grafts. Abundant GFAP⁺ astrocytes were present in the graft regions, and some of them were HUN⁺ (Fig 4F). To exclude the possibility that these GFAP⁺ cells might be RGs, we performed coimmunostaining for GFAP and SOX2, as well as for GFAP and paired box 6 (PAX6) (S4D Fig). The results showed that very few GFAP⁺ cells coexpressed SOX2 or PAX6 (S4D Fig), suggesting that the majority of GFAP⁺ cells in the grafts were astrocytes, some of which were native vOrganoid cells (HUN⁺ cells). Furthermore, we also detected signs of myelination in the grafts upon staining for myelin basic protein (MBP). However, consistent with the observation that very few oligodendrocytes were found in cultured vOrganoids by scRNA-seq, only sparse MPB⁺ signals were observed at the implantation border of the grafts, and no myelination was detected in the core regions (Fig 4G), which is consistent with previous studies [29]. These results suggested that the myelinated fibers in the organoid grafts were primarily derived from the host brain and merely intruded into regions of the implantation border and that barely any came from the organoids in the early stage after implantation.

## Discussion

The neurovasculature, which is the circulatory system in the brain, is a key component of the NSC microenvironment, which provides oxygen and nutrient support to the brain while removing waste metabolites. Proper development and functioning of the CNS relies on mutual cross talk between the nervous and vascular systems [21,24,58]. Brain organoids are promising models for investigating the development of the human brain and the pathomechanism of mental disorders. However, in current culture systems, the ability of oxygen and nutrients to reach the center of organoids is hampered due to impairments in the circulatory system; therefore, cellular necrosis occurs in the center of organoids. This cellular necrosis in organoids, in turn, heavily limits the continuous growth and long-term maintenance of functional cells in organoids. Many studies have tried to generate vOrganoids by adopting various methods. For example, Cakir and colleagues reported a method for engineering human brain organoids with a functional vascular-like system by introducing hESCs ectopically expressing ETV2 into human cerebral organoids[28]. Here, we have established a reproducible system for the generation of vascularized cerebral vOrganoids by coculturing hESCs/hiPSCs with HUVECs, with minimal introduction of extrinsic signals. HUVECs, which are derived from the endothelium of veins from the umbilical cord, have been used extensively to characterize angiogenesis [59,60]. However, HUVECs are different from HBMECs, which are the main ECs in the human brain [39]. Interestingly, we found that coculture with organoids can induce the development of HUVECs towards brain-like ECs to some extent, as indicated by P-gp expression.

In this system, the ECs connected to form a mesh-like and tube-like vascular structure in vOrganoids. In the early culture stage, the vascular structures preferred to be adjacent to the neural progenitors and were located above the VZ/SVZ region. With rapid neurogenesis, the migrating zone and CP expanded while the VZ/SVZ area shrunk. During this process, the vascular structures gradually emerged in the migrating zone and CP in vOrganoids. ECs have been suggested to secrete soluble factors such as vascular endothelial growth factors (VEGFs), angiopoietin-1 (Ang1), and angiopoietin-2 (Ang2), which promote the self-renewal and differentiation of VZ/SVZ progenitors [49,51,52]. Additionally, the VZ/SVZ is more densely vascularized than its neighboring brain regions during development [50]. Several studies have also indicated that neural cells in neighboring areas can participate in feedback mechanisms to regulate the growth and properties of cells in the vasculature [21]. Therefore, our vOrganoid system to a certain extent recapitulates the in vivo interaction between vascular cells and NSCs/progenitors during neurogenesis. In vOrganoids, ECs may secrete essential extrinsic signals that facilitate neurogenesis. The observations of lower hypoxia signals and fewer cleaved CASPASE 3$^+$ cells in vOrganoids than nonvascularized organoids indicate that the vascular systems in vOrganoids may supply even greater levels of oxygen and nutrients for neural progenitors and neurons, although no real blood cells are present in the vascular structure. These factors may promote cell proliferation and differentiation and prevent cell death, which could account for the larger size of and enhanced neurogenesis in vOrganoids. Given the limitations of traditional organoid culture, our vascularized 3D cerebral culture may solve the problem of insufficient oxygen supply and nutrient support to organoids to some extent, thus permitting better growth and functional development of vOrganoids.

vOrganoids mimic cortical development in vivo, and the major cell types in these organoids are RGs, oRGs, IPCs, excitatory neurons, interneurons, astrocytes, and microglia, as detected by both immunostaining and scRNA-seq. In addition, the cell types in cultured vOrganoids and control organoids were both highly similar to those in the human telencephalon at the single-cell transcriptome level [43,61], indicating that vOrganoids could be a potential model for exploring the fundamental questions related to human brain development and neurological diseases. Unfortunately, we did not find clustered ECs in vOrganoids, but this may have been because the dissociation method we used was too mild to break the tight junctions between ECs [62,63]. However, the DEGs between vOrganoids and nonvascularized organoids were enriched in the GO terms of blood vessel morphogenesis, suggesting that there may have been some ECs that were not clustered due to limited cell number caught by scRNA-seq.

Pyramidal excitatory neurons and interneurons in vOrganoids showed an organization that was similar to that observed in the human fetal neocortex. With this classical cytoarchitecture, the neurons gradually matured and formed synaptic connections. We also observed a transition from NMDA receptor–to AMPA receptor–mediated transmission of excitatory synapses in our culture system, indicating that the neurons exhibited a maturation process that contributes to the formation of functional circuits [56]. Together, the results suggest that our vOrganoids derived from coculturing hESCs/hiPSCs with ECs recapitulate human cortical development not only in cell type and cellular organization but also in circuit formation.

A variety of 3D organoid systems have been transplanted in vivo [4,64], and in some cases, they have been used to repair and rescue tissue damage [65], indicating that organoids have the potential to be good resources for cell therapy. Mansour and colleagues proved that the intracerebral implantation of hiPSC-derived brain organoids into immunodeficient mice can develop into a functional vasculature system that significantly rescued the cell death in the grafts and improve the long-term survival of organoid grafts [29]. In their study, there were almost no TUNEL-positive cells detected in the organoid grafts at 50 dpi and 233 dpi.

However, we observed about 10% of cleaved CASPASE 3 cells in the vOrganoid grafts. The differences in the culture and transplantation methods and transplantation location might account for the discrepancy. However, the results from both studies suggest that the growth of blood vessels into the grafts help cell survival. In addition, Cakir and colleagues verified that the vascularized human cortical organoids established in their studies can form a functional vasculature system in vivo by implanting vOrganoids subcutaneously into the hind limbs of immune-deficient mice [28]. However, no reports have demonstrated whether the vascular systems in vOrganoids can connect to the host cerebrovascular systems to form a new functional circulation system when implanted in vivo. In our studies, we explored this question by transplanting vOrganoids into the S1 cortex of NOD-SCID mice. In the blood vessels that were located in the vOrganoid graft regions, both human ECs and mouse endothelial cells were detected. Furthermore, a notable level of blood flow was observed in the vOrganoid grafts in mice by two-photon fluorescence imaging. Therefore, our results illustrate that the vascular system in vOrganoids can recruit host ECs to reconstruct a functional vascular system and enable blood flow into the grafts after implantation. Vascularization is a potential vital factor for the survival of organoids, as it may not only promote cell growth in vitro but also play a role in reconstructing blood vessels after transplantation in vivo. Hence, we speculate that our vascularized culture system may be widely applicable for future 3D organoid transplantation in vivo to improve survival rates and functional reconstruction.

## Materials and methods

### Ethics statement

The animal housing conditions and all experimental procedures in this study were in compliance with the guidelines of the Institutional Animal Care and Use Committee (IACUC) of the Institute of Biophysics, Chinese Academy of Sciences (SYXK2017-22). The human brain tissue collection and research protocols were approved by the Reproductive Study Ethics Committee of Beijing Anzhen Hospital (2014012x) and the Institute of Biophysics (H-W-20131104). The written informed consent was designed as recommended by the ISSCR (International Society for Stem Cell Research) guidelines for fetal tissue donation. The fetal tissue samples were collected after the donor patients signed informed consent documents, which is in strict observance of the legal and institutional ethical regulations from elective pregnancy termination specimens at Beijing Anzhen Hospital, Capital Medical University. All the protocols are in compliance with the "Interim Measures for the Administration of Human Genetic Resources," administered by the Ministry of Science and Technology of China.

### Mice

NOD-SCID–immunodeficient mice were purchased from Charles River Laboratories in China (Vital River, Beijing, China) and used for experiments of organoid implantations. All mice were housed in SPF environments with a 12-hour light–dark schedule and had free access to food and water. All the subjects were not involved in any previous procedures.

### Tissue sample collection

Fetal brains were collected in ice-cold artificial CSF containing 125.0 mM NaCl, 26.0 mM NaHCO3, 2.5 mM KCl, 2.0 mM CaCl2, 1.0 mM MgCl2, and 1.25 mM NaH2PO4 at a pH of 7.4 when oxygenated (95% O2 and 5% CO2). Gestational age was measured in weeks from the first day of the woman's last menstrual cycle to the sample collecting date.

## Three-dimensional vOrganoid culture and differentiation procedure

hESCs (H9 and H3 lines, from ATCC) and iPSCs (AE and LMZ) were maintained on Matrigel-coated 6-well plates (Corning, Corning, NY) and were cultured with Essential 8 Medium (Gibco). On day 0, the hESCs and iPSCs colonies were pretreated for one hour with 20 μM Y27632 (Tocris Bioscience, Bristol, Avon, UK) and were dissociated into single cells by Accutase (STEMCELL Technologies, Vancouver, British Columbia, Canada). HUVECs were cultured in endothelial Cell Growth Medium and dissociated into single cells by TrpLE. The mixture of approximately $3 \times 10^6$ dissociated hESCs and $3 \times 10^5$ HUVECs was resuspended in Knockout Serum Replacement (KSR) medium (Gibco, Waltham, MA) and plated into 96-well U-shaped polystyrene plates (Thermo Fisher, Waltham, MA). The size of embryonic bodies (EBs) is determined by the number of cells seeding into each well of the plate. The KSR medium was prepared as follows: DMEM/F12 (Gibco, Waltham, MA) was supplemented with 20% KSR (Gibco, Waltham, MA), 2 mM Glutamax (Gibco, Waltham, MA), 0.1 mM nonessential amino acids (NEAA; Gibco, Waltham, MA), 0.1 mM beta-mercaptoethanol (Gibco, Waltham, MA), 3 μM endo-IWR1 (Tocris Bioscience, Bristol, Avon, UK), 0.1 μM LDN-193189 (STEMGENT, Beltsville, MD), and 10 μM SB431542 (Tocris Bioscience, Bristol, Avon, UK). On day 18, the self-organized floating EBs were transferred to low-cell-adhesion 6-well plates (Corning Corning, NY) and further cultured in the neural induction medium containing DMEM/F12, 1:100 N2 supplement (Gibco, Waltham, MA), 2 mM Glutamax (Gibco, Waltham, MA), 0.1 mM NEAA (Gibco, Waltham, MA), 55 μM beta-mercaptoethanol (Gibco, Waltham, MA), and 1 μg/mL heparin. After day 35, the free-floating aggregates were transferred to the neurobasal-type differentiation medium supplemented with 1:50 B27 (Gibco, Waltham, MA), 2 mM Glutamax and 0.1 mM NEAA, 0.55 μM beta-mercaptoethanol, 5 μg/mL heparin, 1% Matrigel, 10 ng/mL BDNF, 10 ng/mL GDNF, and 1 μM cAMP (Sigma-Aldrich, St.Louis, MO). The control nonvascularized organoids were generated with the same workflow, except no HUVECs were added. The vascular and cortical structure analyses have been done in organoids from all the cell lines to test the repeatability of our culture procedure; other experiments in the studies were conducted by the H9 cell line.

## Tube formation assay

First, 24-well plates were coated with thawed Matrigel (354230) without introducing air bubbles, and then the plates were plated in a 37°C incubator for at least 30 minutes to allow gelling of the Matrigel. Next, the HUVECs reaching to approximately 80% confluence in a T-25 flask were digested by being incubated in the trypLE solution at 37°C for 3 minutes. After neutralizing the trypLE, 15ml tubes containin HUVECs were centrifuged for 5 minutes at 800$g$, then the supernatant was removed and the cell pellet suspended in the HUVEC medium, EGM (cc-3124, Lonza, Basel, Switzerland). Adjust the volume with HUVEC medium to have $5 \times 10^5$ cells/mL, and plate 200 μL of the cell mixture, which contains 100,000 cells, into each Matrigel-coated well. Then, incubate the 24-well plate overnight at 37°C in a 5% CO2/95% air incubator. The next day, the medium is gently aspirated from each well and is incubated with approximately 500 μL of PFA fixative for 15 minutes. The tube-like structures formed by HUVECs were then immunostained.

## Immunofluorescence

Organoids and tissues were fixed by 4% paraformaldehyde in PBS at 4°C for 2 hours and then dehydrated in 30% sucrose in PBS. The fixed and dehydrated organoids and tissues were embedded and frozen at −80°C in O.C.T. compound, sectioned with Leica CM3050S. Cryosections were subjected to antigen retrieval, pretreated (0.3% Triton X-100 in PBS) and incubated

for a blocking solution (10% normal donkey serum, 0.1% Triton X-100, and 0.2% gelatin in PBS), followed by incubation with the primary antibodies (SOX2, sc17319, Santa Cruz, Dallas, TX; IB4, 217660-100UG, Millipore, Burlington, MA; LAMININ, ab23753, Abcam, Cambridge, Cambridgeshire, UK; TBR2, ab23345, Abcam, Cambridge, Cambridgeshire, UK; CTIP2, NB100-79809, Novus, Littleton, CO; SATB2, ab34735, Abcam, Cambridge, Cambridgeshire, UK; PAX6, 901301, BioLegend, San Diego, CA; MAP2, ab32454, Abcam, Cambridge, Cambridgeshire, UK; SST, MAB354, Millipore, Burlington, MA; CR, 6B3, Swant, Marly, Fribourg, Switzerland; NeuN, MAB377, Millipore, Burlington, MA; SYB2, 102211, SYSY, Goettingen, Lower Saxony, Germany; PSD95, 124003, SYSY, Goettingen, Lower Saxony, Germany; GFAP, 3670s, CST, Danvers, MA; MBP, ab62631, Abcam, Cambridge, Cambridgeshire, UK; CD31, MAB1393Z, Millipore, Burlington, MA; cleaved CASPASE 3, 9664S, CST, Danvers, MA; BRN2, sc-31983, Santa Cruz, Dallas, TX; GABA, LS-C63358-100, LifeSpan, Seattle, WA; AIF1, LS-B2645-50, LifeSpan, Seattle, WA; FOXP2, ab16046, Abcam, Cambridge, Cambridgeshire, UK; HOPX, sc-30216, Santa Cruz, Dallas, TX; P-gp, MA126528, Pierce, Waltham, MA; RELN, MAB5364, Millipore, Burlington, MA; HUN, ab191181, Abcam, Cambridge, Cambridgeshire, UK;) overnight at 4˚C. Immunofluorescence images were acquired with Olympus laser confocal microscope and analyzed with FV10-ASW viewer (Olympus, Tokyo, Japan), ImageJ (NIH), and Photoshop (Adobe).

## Electrophysiology

Cultured vOrganoids were embedded in 3% low-melting agarose in artificial cerebral spinal fluid (ACSF; in mM: 126 NaCl, 3 KCl, 26 $NaHCO_3$, 1.2 $NaH_2PO_4$, 10 D-glucose, 2.4 $CaCl_2$, and 1.3 $MgCl_2$) and sectioned at 200 μm in oxygenated (95% O2 and 5% CO2) ice-cold ACSF with a vibratome (VT1200s, Leica, Wetzlar, Hesse, Germany). The slices were then cultured in a 24-well plate filled with 250 μL/well of neural differentiation medium in an incubator (5% $CO_2$, 37˚C). After a recovery period of at least 24 hours, an individual slice was transferred to a recording chamber and continuously superfused with oxygenated ACSF at a rate of 3–5 mL per minute at 30 ± 1˚C. Whole-cell patch clamp recording was performed on cells of vOrganoid slices. Patch pipettes had a 5–7 MΩ resistance when filled with intracellular solution (in mM: 130 potassium gluconate, 16 KCl, 2 $MgCl_2$, 10 HEPES, 0.2 EGTA, 4 $Na_2$-ATP, 0.4 $Na_3$-GTP, 0.1% Lucifer Yellow, and 0.5% neurobiotin, pH = 7.25, adjusted with KOH). Evoked action potentials were recorded in current-clamp mode using a series of injected currents from −60 pA to 280 pA in increments of 20 pA. Whole-cell currents were recorded in voltage-clamp mode with a basal holding potential of −60 mV followed by stimulating pulses from −80 mV to 60 mV with a step size of 10 mV. The membrane potential was held at −70/0 mV when spontaneous EPSCs/IPSCs were recorded. Dual-patch recording was performed in current-clamp mode. A pair of pulses (1 nA, 2-ms duration, 50-ms interval) were injected into each cell separately. The cells were monitored with a 40× Olympus water-immersion objective lens, a microscope (Olympus, BX51 WI) configured for dodt gradient contrast (DGC), and a camera (Andor, iXon3, Belfast, County antrim, UK). Stimulus delivery and data acquisition were conducted with a multiclamp 700B amplifier and a Digidata 1440A (Molecular Devices, San Jose, CA), which were controlled by Clampex 10. The slices were fixed after patch clamp recording. Staining with fluorescein streptavidin (SA-5001, 1:500, Vector, Burlingame, CA) or Texas Red streptavidin (SA-5006, 1:500, Vector, Burlingame, CA) was performed to visualize the morphology of cells.

## Calcium imaging

Three microliters of dye solution that contained 50 μg Fluo-4 AM (Life Technologies), 50 μL DMSO, and 200 μg Pluronic F-127 (Sigma) was applied to the surface of each individual slice.

After incubation for 30 minutes at 37°C, the slice was transferred to a recording chamber and continuously superfused with oxygenated ACSF (in mM: 126 NaCl, 3 KCl, 26 NaHCO3, 1.2 NaH2PO4, 10 D-glucose, 2.4 CaCl2, and 1.3 MgCl2) at a rate of 3–5 mL per minute at 30 ± 1°C. The slices were washed 30 minutes before imaging. Calcium imaging was acquired at 5 Hz using a camera (Andor iXon3) with a FITC filter set (Ex: 475/35 nm, Em: 530/43 nm) on a BX51WI microscope (Olympus). Data analysis was performed with ImageJ. The ROIs were selected manually, and the mean fluorescence (F) was calculated for each frame. The fluorescence changes over time was calculated as follows: $\Delta F = (F − F_{basal})/F_{background}$, in which $F_{basal}$ was the lowest mean fluorescence value during imaging, and $F_{background}$ was the average mean fluorescence across all frames. Since 10 minutes before imaging, TTX (1 μM), CNQX (20 μM), APV (100 μM), and BMI (20 μM) were added by bath application. The slices were rinsed for 30 minutes with ACSF after drug treatments. Different from other drugs, glutamate (100 μM) was added by bath application during imaging.

## Implantation of cerebral organoid into mice S1 cortex

Before implantation, cerebral organoids had been cultured for 60 days, and strict screening was performed by bright field (BF) microscopy to select the organoids with the appropriate size and displayed without massive cyst formation. Immune-deficient NOD-SCID mice, aged 8 weeks, were used in our studies. Mice were anesthetized by intraperitoneal injection of avertin. The heads of animals were fixed in a stereotactic frame. The fur above the skull was removed and skin was cut. An approximate 3-mm diameter craniotomy was performed by polishing the skull; the underlying dura mater was removed and a cavity was made by aspiration with a blunt-tip needle attached to a vacuum line. The aspirative lesion was made unilaterally in the region of the S1cortex. Sterile ACSF was used to irrigate the lesion and keep it free of blood throughout the surgery, and a piece of Gelfoam (Pfizer, New York, NY) was used to slow the bleeding and absorb the excess blood. Molten 3% low-melting agarose was dropped in the implantation regions to immobilize the organoid grafts as agarose congealed quickly. Adhesive glue was used to seal the border of implanted organoid grafts. The wound was closed with sutures. Following completion of the surgery, penicillin streptomycin combination was administrated for inflammation and analgesic relief. The mice were then returned into home cages to recover.

## Two-photon imaging

For imaging of blood flow in organoid grafts, mice were tail intravenously injected FITC-dextran. The mouse was fixed on the recording setup, with an isoflurane-oxygen mixture of 0.5%–1% (v/v). The in vivo imaging of blood flow was done with a 2-photon laser scanning microscope. The recording chamber was perfused with normal ACSF containing 126 mM NaCl, 3 mM KCl, 1.2 mM NaH$_2$PO$_4$, 2.4 mM CaCl$_2$, 1.3 mM MgCl$_2$, 26 mM NaHCO$_3$, and 10 mM D-glucose (pH 7.4 when bubbled with 95% oxygen and 5% CO$_2$). The temperature of the mouse was kept at approximately 37°C throughout the experiment.

## Single-cell dissociation, and library construction

The organoids with and without HUVECs at 65 days and 100 days, respectively, were cut into small pieces and dissociated into single-cell suspensions by using a papain-based dissociation protocol (hibernate E medium with 1 mg/mL papain [Sigma, St.Louis, MO] at 37°C on a thermo cycler at 500$g$ for 15–20 minutes). Single cells were suspended in 0.04% BSA/PBS at the proper concentration to generate cDNA libraries with Single Cell 3′ Reagent Kits, according to the manufacturer's protocol. Briefly, after the cDNA amplification, enzymatic

fragmentation and size selection were performed to optimize the cDNA size. P5, P7, an index sample, and R2 (read 2 primer sequence) were added to each selected cDNA during end repair and adaptor ligation. P5 and P7 primers were used in Illumina bridge amplification of the cDNA (http://10xgenomics.com). Finally, the library was processed on the Illumina platform for sequencing with 150-bp paired-end reads.

## Single-cell RNA-seq data preprocessing

Raw sequencing data were processed using Cell Ranger analysis pipeline 2.1.1 with default parameters. Reads were aligned to human reference genome (GRCh38). Cell Ranger output "filtered gene-barcoded" count matrix was loaded into Seurat 3.1.0 [66] for downstream analysis. We excluded poor-quality cells based on the following criterion: nFeature_RNA > 200 or nFeature_RNA < 6,000, mitochondrial gene percentage < 10%. Cells with percentage of hemoglobin reads > 1% ('HBA1', 'HBA2', 'HBB', 'HBD', 'HBE1', 'HBG1', 'HBG2', 'HBM', 'HBM', 'HBQ1', 'HBZ') were discarded as well. In total, 57,180 cells remained for subsequent analysis. Batch effect correction was performed with function fastMNN from R package batchelor [67] by considering each sample as a batch.

## Dimensionality reduction, cell clustering, and cell type identification

We performed principal component analysis (PCA) with Seurat function RunPCA [66] and selected the top 20 statistically significant PCs for clustering by using function FindClusters with resolution parameter set to 2. Known markers *SOX2*, *FAM107A*, *MKI67*, *EOMES*, *DCX*, *NEUROD2*, *GAD1*, *AIF1*, *AQP4*, *OLIG1*, and *RSPO2* were used to name the major cell types RG, oRG, cell cycle, IPC, immature neuron, excitatory neuron, interneuron, microglia, astrocyte, oligodendrocyte and choroid plexus, respectively. UMAP was employed for visualization of our data in 3D coordinates, generated by the RunUMAP [66] function. Three-dimensional UMAP plots were generated using function plot3d from R package rgl.

## Identification of DEGs among clusters

Differential gene expression analysis among clusters was performed with Seurat function FindAllMarkers by setting parameter "only pos = TRUE" [66]. Genes with adjusted *p*-values <0.05 were selected as DEGs.

## Ordering cells along pseudo-maturation trajectory and pseudo-time trajectory

To order cells along a pseudo-maturation trajectory, we applied the approach introduced in Petropoulos's [68] and Mayer's [69] studies. Following these approaches, we performed dimensionality reduction analysis with Seurat function RunPCA based on the expression of variable genes, followed by fitting a principal curve through the subspace spanned by the first three principal components (PCs). The maturation score for a cell was defined as the arc-length from the beginning of the fitted curve to the points where the cell was fitted on. Then, to assign the curve a starting point, we correlated the PCs with maturation scores and defined the beginning of the curve so that the expression of gene SOX2 is negatively correlated with the maturation score. Once each cell was projected onto the principal curve and given a maturation score, we computed the density of cells along the maturation trajectory for organoids and vOrganoids with R function hist by setting freq = False and fitted curves through them.

Besides a pseudo-maturation score, we also constructed a pseudo-time trajectory with R package monocle3 [44–46]. We first imported the gene expression matrix into monocle,

followed by dimensionality reduction analysis with function reduce_dimension, which generated a UMAP graph. We then ran function learn_graph to learn the generated trajectory graph. For ordering cells, we applied the function order_cells by manually selecting the cell cluster that annotated as RG (the cluster that had a higher expression level of the known radial glial marker gene SOX2) as root. After each cell was assigned a pseudo-time value, as described above for maturation trajectory, we computed the density of cells along the pseudo-time trajectory and fitted curves on data.

## Mapping organoids cell types to fetal human cortex cell types

To compare cell-type classification between organoid and human cortex datasets, we downloaded count matrix from the previous published work (Nowakowski and colleagues, 2017). Cell types, as assigned to each cell in the previous work, were merged into more general cell types (for example, cells with identity of nIN and IN-CTX were aggregated into a general cell type, Interneuron; EN-PFC and EN-V1 were merged into cell type Excitatory neuron). The Seurat integration approach is committed to identify the homogeneous cell states across different datasets based on dectecting shared sources. Thus, the function of Seurat integration was adopted to integrate the datasets of organoids and human in our studies., and so to combine First, we selected genes for the integrated analysis by function SelectIntegrationFeatures with default parameters. As described in Stuart's study (Stuart and colleagues, 2019), this function ranked highly variable genes that individually identified in the organoid and fetal human cortex datasets by examining the number of datasets in which they were independently identified as highly variable genes. Next, we identified correspondences among the two datasets using function FindIntegrationAnchors based on the genes selected, as described above [66], and then integrated them into a combined dataset with function IntegrateData, which holds the integrated (or 'batch-corrected') gene expression matrix for all cells from the combined dataset. To compute the relations between cell types among organoid and human datasets, we imported the integrated matrix into PAGA and constructed an Anndata data structure followed by running PCA with function sc.tl.pca. Next, we computed the neighborhood graph with function sc.pp.neighbors and constructed the PAGA graph with function sc.tl.paga [70,71]. (the width of edges was scaled with the strength of relations; the size of nodes was scaled with the number of cells of each cell type). To access a general visualization of the distribution of cells, a force-directed graph (ForceAtlas2 layout) of the combined dataset was computed with function sc.tl.draw_graph [70,71].

## GO enrichment analysis

GO enrichment analysis was performed with Metascape [72] by importing the DEGs between H9-d65 and HUVEC-d65 cells and visualized via heatmap. Only GO terms with an adjusted $p$-value $<0.05$ (Benjamini-Hochberg correction for multiple testing) were considered.

## Quantification and statistical analysis

All data were represented as the mean ± SEM. The quantification graphs were made using GraphPad Prism software. The sample size ($n$) for each analysis can be found in the figure legends.

## Data resources

The accession number for the RNA sequencing data reported in this paper is GEO: GSE131094.

## Supporting information

**S1 Checklist. MDAR checklist.** MDAR, Materials Design Analysis Reporting.
(DOCX)

**S1 Fig. The vascular system in vOrganoids promotes cell growth and reduces cell apoptosis.** (A) Representative figure showing the tube formation of HUVECs on matrigel. Scale bar, 200 μm. (B) Representative immunofluorescence staining figure for CD31, IB4, and SOX2 to illustrate that tube-like vascular systems were formed in vOrganoids at 45 days. Scale bar, 50 μm. (C) Representative immunostaining figure for IB4 in the human cortical cryosections at GW12 to show the distribution of blood vessels in the human cortex. Scale bar, 100 μm. (D) Representative BF and immunohistochemical images of vOrganoids derived from hESC-3 and hESC-9 lines. BF images showed on the left and immunohistochemical images for vessel (IB4, green) and progenitor (SOX2, red) showed on the right. Scale bar, 100 μm (left panels), 50 μm (right panels). (E) Representative BF and immunohistochemical images of vOrganoids derived from iPSCs-AE and iPSCs-LMZ cell lines. BF images showed on the left and immunohistochemical images for vessel (IB4, green) and progenitor (SOX2, red) showed on the right. Scale bar, 100 μm (left panels), 50 μm (right panels). (F) Representative immunofluorescence staining figure for cleaved CASPASE 3, IB4 (upper) and HIF1α, IB4 (lower) in the control nonvascularized organoids and vOrganoids at d115. Scale bar, 500 μm. (G) Quantification of the percentages of cleaved CASPASE $3^+$ cells (left) and HIF1α$^+$ cells (right) within all cells (DAPI$^+$) in the control organoids and vOrganoids at d115, respectively. For cleaved CASPASE 3, $n$ = 5, 5 slices of the control organoids and vOrganoids from three independent experiments. For HIF1α, $n$ = 5, 5 slices of control and vOrganoids in three independent experiments. Data are represented as mean ± SEM, independent samples $t$ test, $^{***}p < 0.001$. (H) The diameters of organoids and vOrganoids generated from H9 at day 7, day 31, day 52, day 70 and day 98, respectively. $n$ = 11, 11, 11, 11, 11 for day 7, day 31, day 52, d 70, and day 98, respectively. Data are represented as mean ± SEM, two-way ANOVA analysis, $^{***}p < 0.001$. (I) Representative images showing the distribution of PAX6$^+$ progenitors in the organoids with or without HUVECs (IB4, red). Scale bar, 50 μm. (J-K) Quantification of the percentages of PAX6$^+$ cell within all cells (DAPI$^+$) in VZ/SVZ (J) and of the thickness of PAX6$^+$ region (K) in control organoids and vOrganoids, respectively. For (J), $n$ = 3, 3 slices from control organoids and vOrganoids in three independent experiments, respectively. For (K), $n$ = 4, 5 slices from control organoids and vOrganoids in three independent experiments, respectively. Data are represented as mean ± SEM, independent samples $t$ test, $^*p < 0.05$. (L) Representative immunofluorescence staining figure for P-gp and IB4 in the human cortical slices at GW12. Scale bar, 100 um. (M) Representative immunofluorescence staining figure for P-gp and IB4 in the tube-like structure formed by HUVECs. Scale bar, 100 um. (N) Representative immunofluorescence staining figure for P-gp and IB4 in the vOrganoids at d83. The signals of P-gp were colocalized with IB4 to a great degree. Scale bar, 200 μm. (O) Representative immunofluorescence staining figure for SATB2 and FOXP2 in the human fetal cortex at GW23 to show the human cortical lamination. Scale bar, 50 μm. (P) Cryosections of vOrganoids were immunostained for the progenitor (PAX6) and layer-specific cortical neuron marker (SATB2) at 65 days. Scale bar, 50 μm. Representative figure was showed. (Q) Cryosections of vOrganoids were immunostained for the layer-specific cortical neuron markers, RELN and TBR1, at 65 days. Scale bar, 100 μm. Representative figure was showed. The numerical data underlying this figure can be found in the S1G, S1H, S1J, S1K Fig sheets of S1 Data. BF, bright field; CD31, platelet and endothelial cell adhesion molecule 1; FOXP2, forkhead box P2; GW, gestational week; hESC, human embryonic stem cell; HIF1α, hypoxia inducible factor 1 subunit alpha; HUVEC, human umbilical vein endothelial cell; IB4, isolectin I-B4; iPSC, induced pluripotent

stem cell; P-gp, P-glycoprotein; PAX6, paired box 6; RELN, reelin; SATB2, SATB homeobox 2; SOX2, SRY-box transcription factor 2; TBR1, T-box brain transcription factor 1; vOrganoid, vascularized organoid; VZ/SVZ, ventricular zone/subventricular zone.
(TIF)

**S2 Fig. scRNA-seq of organoids with and without HUVECs.** (A) The cell distributions of each sample of control organoid and vOrganoid were showed in the UMAP plots. As for the control organoids and vOrganoids at each time point, three independent batches of experiments were performed. And in total, 12 samples were included in the studies. Each sample was colored differently in the UMAP plot. (B) Quality control for samples: each dot represents a single cell. Cells with mitochondrial gene percentage >10% (left panel), as well as gene number per cell (nGene) <200 and >6,000 (right panel), were discard in the following analysis. (C) The expression of known gene markers was visualized by UMAP plots and was colored by the expression level (red, high; gray, low). (D) Heatmap showing the expression of DEGs across clusters. Some canonical marker genes were labeled. (E) The transcriptomic correlations between the cell types of organoids and vOrganoid were visualized via heatmap (correlation coefficient: yellow, high; purple, low). (F) The expression of the well-known gene markers in different cell types is showed in the PAGA plots. AQP4, DCX, SOX2, RSPO2, FAM107A, and MKI67 are the specific genes for astrocytes, immature neurons, RG/oRG, choroid plexus, oRG, and cell cycle active cells, respectively. Nodes are colored according to the gene expression levels (light pink, low; dark red, high). (G-H) After being integrated, the cell distributions of organoids (G) and human fetal telencephalon (H) were visualized in a forced-directed graph separately. Each dot represents a single cell and is colored according to the cell types. The width of edges is scaled with the cell–cell connectivity. (I-J) After being integrated, the cell distributions of vOrganoids (I) and human fetal telencephalon (J) were visualized in a forced-directed graph separately. Each dot represents a single cell and is colored according to the cell types. The width of edges is scaled with the cell–cell connectivity. (K-L) The correlations between the cell types of organoids and human fetal telencephalon (K) as well as those between the cell types of vOrganoids and human fetal telencephalon (L) are displayed in the PAGA graph. The cell types in the organoids and human fetal cortex datasets were denoted by the solid and hollow dots, respectively. And the same cell types in two datasets were colored identically. The size of dots was scaled with the cell numbers; width of edges was scaled with the connectivity between cell types. (M-N) The transcriptomic correlations between the cell types of control organoids (M) or vOrganoids (N) and human fetal telencephalon were visualized via heatmap (correlation coefficient: red, high; blue, low). (O) The composition of cell types in the control organoids and vOrganoids at day 65 was illustrated in the bar graph. The numerical as well as metadata underlying this figure can be found in the S2A, S2B, S2C, S2D, S2E, S2G, S2H, S2I, S2J, S2M and S2N Fig sheets of S1 Data. Ast, astrocyte; cell cycle, cell cycle active cell; Ch, choroid plexus; DEG, differentially expressed gene; Endo, endothelial cell; ExN, excitatory neuron; HUVEC, human umbilical vein endothelial cell; ImN, immature neuron; IN, interneuron; IPC, intermediate progenitor cell; MGE, medial ganglionic eminence; MGE div, MGE dividing cell; Mic, microglia; Mural, mural cell; OPC, oligodendrocytes progenitor cell; oRG, outer radial glia; PAGA, partition-based graph abstraction; RG, radial glia cell; scRNA-seq, single-cell RNA sequencing; UMAP, Uniform Mainfold Approximation and Projection; vOrganoid, vascularized organoid.
(TIF)

**S3 Fig. Functional circuits were built in vOrganoids progressively.** (A) Diameters of somata of recorded cells from control organoids and vOrganoids at day 60, day 80, and day 90. Open circles indicate data from individual cells. Filled circles indicate the mean value. Data are

represented as mean ± SEM, $p$ = 0.6026, 0.3494, and 0.7491 for day 60, day 80, and day 90, two-sample $t$ test, $^*p < 0.05$, $^{**}p < 0.01$, $^{***}p < 0.001$. (B) Depth of somata of recorded cells from organoids and vOrganoids at day 60, day 80, and day 90. Open circles indicate data from individual cells. Filled circles indicate the mean value. Data are represented as mean ± SEM, $p$ = 0.5799, 0.5248, and 0.8651 for day 60, day 80, and day 90 two-sample $t$ test, $^*p < 0.05$, $^{**}p < 0.01$, $^{***}p < 0.001$. (C) Representative current responses evoked by a series of voltage steps (from −80 mV to +60 mV, in steps of 10 mV) from organoid (upper) and vOrganoid (lower) cells at day 60, day 80, and day 90. (D) Representative evoked action potentials of cell from organoid and vOrganoid at day 90. The amplitudes of injected currents changed from −60 pA to 280 pA in steps of 20 pA. (E) The percentage of firing cells under current stimulations with different amplitudes. The insertion indicates the evoked action potential of cells from organoids (black) and vOrganoids (red). The scale bars indicate 5 ms and 20 mV. The arrow indicates −20 mV. $p$ = 0.0602, 0.0000025, and 0.00012 for day 60, day 80, and day 90. Data are represented as mean ± SEM, paired-sample $t$ test, $^*p < 0.05$, $^{**}p < 0.01$, $^{***}p < 0.001$, $^{****}p < 0.0001$. (F) Resting membrane potentials of recorded cells from organoids and vOrganoids at day 60, day 80, and day 90. Open circles indicate data from individual cells. Filled circles indicate the mean value. Data are represented as mean ± SEM, $p$ = 0.9842, 0.00059, and 0.4102 for day 60, day 80, and day 90, two-sample $t$ test, $^*p < 0.05$, $^{**}p < 0.01$, $^{***}p < 0.001$. (G) Capacitance of recorded cells from organoids and vOrganoids at day 60, day 80, and day 90. Open circles indicate data from individual cells. Filled circles indicate the mean value. Data are represented as mean ± SEM. $p$ = 0.3059, 0.0424, and 0.0075 for day 60, day 80, and day 90, two-sample $t$ test, $^*p < 0.05$, $^{**}p < 0.01$, $^{***}p < 0.001$. (H) An example of spontaneous action potentials recorded from cells of the vOrganoids at day 80. (I) Postsynaptic currents were sensitive to BMI. Representative IPSC event highlighted by a yellow bar displayed at the right panel. (J) Representative staining figure of classical neuron marker NeuN (green) and MAP2 (red) in the vOrganoid of day 210. Scale bar, 100 μm. (K) An example of whole-cell configuration in the vOrganoids of day 210 under DGC. Scale bar, 200 μm. (L) Representative traces of membrane currents elicited by a series of depolarizing pulses (from −80 mV to +60 mV, in steps of 10 mV) from a neuron in the vOrganoid of day 210. (M) Spontaneous EPSCs and IPSCs recorded from the same cells in (L). Yellow bars indicate representative events, which are shown at right. (N-O) Intracellular spontaneous calcium fluctuations were measured before and after the application of 1 μM TTX (N) or 100 μM glutamate (O) in the vOrganoids. Arrows mark the time of addition of glutamate. Scale bar, 100 μm in (N), 50 μm in (O). (P) Injected cells and the cells that coupled with them through gap junctions were visualized by staining of neurobiotin. Scale bar, 20 μm. The numerical data underlying this figure can be found in the S3A, S3B, S3E, S3F, and S3G Fig sheets of S1 Data. BMI, bicuculline methiodide; DGC, dodt gradient contrast; EPSC, excitatory postsynaptic current; IPSC, inhibitory postsynaptic current; MAP2, microtubule associated protein 2; NeuN, RNA binding fox-1 homolog 3; ROI, region of interest; sEPSC, spontaneous excitatory postsynaptic current; sIPSC, spontaneous inhibitory postsynaptic current; TTX, tetrodotoxin; vOrganoid, vascularized organoid. (TIF)

**S4 Fig. The vascular system formation in vOrganoids grafts. (A)** Representative immunofluorescence staining figure for cleaved CASPASE 3, IB4 in cultured vOrganoids at d120 and for cleaved CASPASE 3, HUN in the control nonvascularized organoid grafts and the vOrganoid grafts at 60 dpi. Scale bar, 50 μm. **(B)** Quantification of the percentages of cleaved CASPASE $3^+$ cells within all cells (DAPI$^+$) in the cultured vOrganoids at d120, nonvascularized and vOrganoid grafts at 60 dpi, respectively. $N$ = 4, 4, 4 samples for the vOrganoids, the nonvascularized organoid grafts, and vOrganoid grafts from three independent experiments, respectively.

Data are represented as mean ± SEM, one-way ANOVA, $^{**}p < 0.01$, $^{***}p < 0.001$. The numerical data underlying this figure can be found in the S4B Fig sheet of S1 Data. **(C)** Representative immunofluorescence staining figure for LAMININ and IB4 in the cryosections of host brains contained the control organoid and vOrganoid grafts at 30 dpi. The immunostaining of LAMININ is individually displayed in the right panels. The boundary between host and graft is outlined by dashed lines. And the arrows point out the healthy-looking blood vessels in the vOrganoid grafts. Scale bar, 100 μm. **(D)** Representative immunofluorescence staining figure for GFAP/SOX2 and GFAP/PAX6 in the vOrganoid grafts at 60 dpi. Scale bar, 50 μm. CAS3, CASPASE 3; dpi, days postimplantation; GFAP, glial fibrillary acidic protein; HUN, human nuclear; IB4, isolectin I-B4; PAX6, paired box 6; SOX2, SRY-box transcription factor 2; vOrganoid, vascularized organoid.
(TIF)

**S1 Movie. Movie shows the staining of LAMININ (red) and IB4 (green) displayed in 3D, rotating around x-, y-, and z-axes.** IB4, isolectin I-B4.
(AVI)

**S2 Movie. Movie shows the 3D reconstruction of LAMININ-positive vascular structure rotating around the y-axis.**
(AVI)

**S3 Movie. Movie shows the functional blood flow in vOrganoids grafts.** vOrganoid, vascularized organoid.
(AVI)

**S4 Movie. Movie shows the 3D reconstruction of dextran-positive blood vessels in vOrganoids grafts.** vOrganoid, vascularized organoid.
(MOV)

**S1 Table. Samples and genes related to scRNA-seq data.** The spreadsheets include the sample information of scRNA-seq, DEGs of major cell types of organoids, and GO terms of the DEGs between progenitor cells. DEG, differentially expressed gene; GO, Gene Ontology; scRNA-seq, single-cell RNA sequencing.
(XLSX)

**S1 Data. The numerical data used in all figures are included in S1 Data.** Excel spreadsheet containing, in separate sheets, the underlying numerical data and statistical analysis for figure panels 1M, 2A, 2B, 2C, 2I, 3A, 3B, 3D, 3F, 3I, S1G, S1H, S1J, S1K, S2A, S2B, S2C, S2D, S2E, S2G, S2H, S2I, S2J, S2M, S2N, S3A, S3B, S3E, S3F, S3G, and S4B.
(XLSX)

## Acknowledgments

The authors thank Junying Jia for the technical support of FACS and Junjing Zhang for management of laboratory animals. We thank all the members of the Wang lab for discussion and technical assistance.

## Author Contributions

**Conceptualization:** Qian Wu, Xiaoqun Wang.

**Data curation:** Yingchao Shi, Le Sun, Mengdi Wang, Suijuan Zhong, Rui Li, Peng Li, Lijie Guo, Ai Fang, Ruiguo Chen, Qian Wu.

**Formal analysis:** Yingchao Shi, Le Sun, Mengdi Wang, Jianwei Liu, Suijuan Zhong, Lijie Guo, Qian Wu.

**Funding acquisition:** Qian Wu, Xiaoqun Wang.

**Methodology:** Mengdi Wang.

**Supervision:** Woo-Ping Ge, Qian Wu, Xiaoqun Wang.

**Validation:** Yingchao Shi, Lijie Guo.

**Writing – original draft:** Yingchao Shi, Mengdi Wang, Qian Wu, Xiaoqun Wang.

**Writing – review & editing:** Yingchao Shi, Le Sun, Mengdi Wang, Jianwei Liu, Suijuan Zhong, Rui Li, Peng Li, Lijie Guo, Ai Fang, Ruiguo Chen, Woo-Ping Ge, Qian Wu, Xiaoqun Wang.

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
