## [Editor Report · Decision Letter 0]

13 Jul 2019

Dear Dr Wang, 

Thank you for submitting your manuscript entitled "Vascularized human cortical organoids model cortical development in vivo" for consideration as a Methods and Resources by PLOS Biology.

Your manuscript has now been evaluated by the PLOS Biology editorial staff as well as by an academic editor with relevant expertise and I am writing to let you know that we would like to send your submission out for external peer review.

**Important**: Please also see below for further information regarding completing the MDAR reporting checklist. The checklist can be accessed here: https://plos.io/MDARChecklist

Please re-submit your manuscript and the checklist, within two working days, i.e. by Jul 15 2019 11:59PM.

Kind regards,

Di Jiang

PLOS Biology

INFORMATION REGARDING THE REPORTING CHECKLIST:

PLOS Biology is pleased to support the "minimum reporting standards in the life sciences" initiative (https://osf.io/preprints/metaarxiv/9sm4x/). This effort brings together a number of leading journals and reproducibility experts to develop minimum expectations for reporting information about Materials (including data and code), Design, Analysis and Reporting (MDAR) in published papers. We believe broad alignment on these standards will be to the benefit of authors, reviewers, journals and the wider research community and will help drive better practise in publishing reproducible research. 

We are therefore participating in a community pilot involving a small number of life science journals to test the MDAR checklist. The checklist is intended to help authors, reviewers and editors adopt and implement the minimum reporting framework. 

IMPORTANT: We have chosen your manuscript to participate in this trial. The relevant documents can be located here:

MDAR reporting checklist (to be filled in by you): https://plos.io/MDARChecklist

**We strongly encourage you to complete the MDAR reporting checklist and return it to us with your full submission, as described above. We would also be very grateful if you could complete this author survey:

https://forms.gle/seEgCrDtM6GLKFGQA

Additional background information:

Interpreting the MDAR Framework: https://plos.io/MDARFramework

Please note that your completed checklist and survey will be shared with the minimum reporting standards working group. However, the working group will not be provided with access to the manuscript or any other confidential information including author identities, manuscript titles or abstracts. Feedback from this process will be used to consider next steps, which might include revisions to the content of the checklist. Data and materials from this initial trial will be publicly shared in September 2019. Data will only be provided in aggregate form and will not be parsed by individual article or by journal, so as to respect the confidentiality of responses. 

Please treat the checklist and elaboration as confidential as public release is planned for September 2019.

We would be grateful for any feedback you may have.

---

## [Decision Letter · Decision Letter 1]

9 Aug 2019

Dear Dr Wang,

Thank you very much for submitting your manuscript "Vascularized human cortical organoids model cortical development in vivo" for consideration as a Methods and Resources at PLOS Biology. Your manuscript has been evaluated by the PLOS Biology editors, an Academic Editor with relevant expertise, and by three independent reviewers.

In light of the reviews (below), we will not be able to accept the current version of the manuscript, but we would welcome resubmission of a much-revised version that address all of the reviewers' comments. It is particularly important to note that the reviewers aren't yet convinced that your work represents a substantial improvement relative to current methods. We must be convinced by your revision for further consideration in our journal. We cannot make any decision about publication until we have seen the revised manuscript and your response to the reviewers' comments. Your revised manuscript is also likely to be sent for further evaluation by the reviewers.

Your revisions should address the specific points made by each reviewer. Please submit a file detailing your responses to the editorial requests and a point-by-point response to all of the reviewers' comments that indicates the changes you have made to the manuscript. In addition to a clean copy of the manuscript, please upload a 'track-changes' version of your manuscript that specifies the edits made. This should be uploaded as a "Related" file type. You should also cite any additional relevant literature that has been published since the original submission and mention any additional citations in your response. 

Before you revise your manuscript, please review the following PLOS policy and formatting requirements checklist PDF: http://journals.plos.org/plosbiology/s/file?id=9411/plos-biology-formatting-checklist.pdf. It is helpful if you format your revision according to our requirements - should your paper subsequently be accepted, this will save time at the acceptance stage.

Please note that as a condition of publication PLOS' data policy (http://journals.plos.org/plosbiology/s/data-availability) requires that you make available all data used to draw the conclusions arrived at in your manuscript. If you have not already done so, you must include any data used in your manuscript either in appropriate repositories, within the body of the manuscript, or as supporting information (N.B. this includes any numerical values that were used to generate graphs, histograms etc.). For an example see here: http://www.plosbiology.org/article/info%3Adoi%2F10.1371%2Fjournal.pbio.1001908#s5.

For manuscripts submitted on or after 1st July 2019, we require the original, uncropped and minimally adjusted images supporting all blot and gel results reported in an article's figures or Supporting Information files. We will require these files before a manuscript can be accepted so please prepare them now, if you have not already uploaded them. Please carefully read our guidelines for how to prepare and upload this data: https://journals.plos.org/plosbiology/s/figures#loc-blot-and-gel-reporting-requirements.

Upon resubmission, the editors will assess your revision and if the editors and Academic Editor feel that the revised manuscript remains appropriate for the journal, we will send the manuscript for re-review. We aim to consult the same Academic Editor and reviewers for revised manuscripts but may consult others if needed.

We expect to receive your revised manuscript within three months. Please email us (plosbiology@plos.org) to discuss this if you have any questions or concerns, or would like to request an extension. At this stage, your manuscript remains formally under active consideration at our journal; please notify us by email if you do not wish to submit a revision and instead wish to pursue publication elsewhere, so that we may end consideration of the manuscript at PLOS Biology.

When you are ready to submit a revised version of your manuscript, please go to https://www.editorialmanager.com/pbiology/ and log in as an Author. Click the link labelled 'Submissions Needing Revision' where you will find your submission record. 

Sincerely,

Di Jiang

PLOS Biology

Reviewer remarks:

Reviewer #1: In this study by Shi et al, the authors generate vascularized cerebral organoids (vOrganoids) from human embryonic stem cells mixed with human endothelial cells in vitro. These vOrganoids display the appropriate neural and endothelial cell markers, structural organization, and functionality typically seen in other organoid studies. The authors implant vOrganoids intracerebrally into mouse brain and observe functional connectivity of vOrganoid vessels with the endogenous vasculature indicated by live two-photon imaging of bloodflow and 3-D reconstruction of vessel morphology. Furthermore, the neurons within vOrganoids are able to mature electrophysiologically as shown by patch-clamp electrophysiology and calcium imaging; these data indicate the presence of both excitatory and inhibitory post-synaptic currents as well as gap junction coupling between neurons suggesting the presence of both chemical and electrical synapses in vOrganoids. These data along with the comprehensive single Cell RNA seq data are valuable to other researchers in the organoid field and support the utility of this system. Furthermore, the authors demonstrate that cerebral implantation can support the long-term health and maturation of organoids to better mimic in vivo development. This study is well-designed and rigorous in its evaluation of vOrganoids and their potential for future research. The data provided are in high quality. The study will be of interest to the field of neuroscience and stem cell biology, as well as disease modeling using pluripotent stem cells. Nevertheless, I have the following suggestions may help further improve the study. 

1. In the first paragraph on page 7 the authors state “….which may account for the well-developed vascularized organoids with reduced cell death.” Can the authors please provide some evidence? Using TUNEL or caspase staining for example.

2. Do the authors have any evidence of synaptic integration between the grafted vOrganoid and mouse brain? Such as colocalization of synaptic markers or via electrophysiology. I recognize that this will be a time-consuming experiment. If sections are available, some immunohistochemistry with high-resolution confocal imaging analyses could be helpful.

Minor suggestions. 

1. “vOrganoid” needs to be defined in the Abstract section.

2. Usages of “GABAA receptor” and “GABAAR” ; “NMDA receptor” and “NMDAR” ; “AMPA receptor” and “AMPAR” should have some consistency. 

3. Figure 1 panel I is not bolded like the other panel letters.

4. What is the relevance of the (#) in the top left of Figure 1 panels I, J, K, L? Statistics should be better defined in figure legend. 

5. While infrequent, this study does have some grammatical/syntax errors. I recommend that the authors stringently go over the manuscript and give it to others with a strong command of English to fix these errors.

6. On page 4 the authors state “vRG, oRG, and IPC” acronyms without first defining these for the reader

Reviewer #2: Shi et al. present a method to generate vascularized human cortical organoids (vhCOs) by co-culturing human umbilical vein endothelial cells (HUVECs) with human embryonic stem cells (hESCs). They also graft vhCOs into a mouse cortex and demonstrate that the mouse vasculature and human vascular cells join in forming blood vessels. This manuscript complements the existing literature regarding vascularized cerebral organoids and grafting the organoids into the rodent brain, including Pham et al., 2018 and Mansour et al, 2018. While the technique presented in this manuscript may be a viable way to improve on cerebral organoid technologies, the main novelty is the usage of HUVECs instead of stem cell-derived endothelial cells (Pham et al., 2018) and is therefore somewhat limited. In addition, some of the conclusions in the manuscript are somewhat tenous. Firstly, the claim that the vhCOs present an advantage over non-vascularized ones was not substantiated. Secondly, the claim that this method is highly reproducible was not supported by the data. The study is also somewhat lacking in rigor and statistical analysis. The major concerns are as follows:

1) One of the main arguments of the manuscript is that the vascularized organoids provide an advantage over non-vascularized ones. The authors found that vascularized organoids are slightly larger in Fig S1E, but this is not inherently an advantage. Moreover, they claimed that there is a relatively small proportion of Caspase-3 positive apoptotic cells in grafted vhCOs (Fig S4A), but there is no control group for comparison. More importantly the authors did not present data on Caspase-3 or TUNEL staining of ungrafted vhCOs grown in parallel to non-vascularized ones to demonstrate a decrease in cell death that would support their claim of reduced cell death. Also, no markers for hypoxia were included to support the claim of better oxygenation.

2) The authors also claimed that based on single cell RNAseq data, the vhCOs “exhibited microenvironments to promote neurogenesis and neuronal maturation that resembled in vivo processes.” While the scRNAseq showed the presence of neurogenesis and neuronal maturation, it cannot demonstrate the presence of microenvironments or stem cell niches. The authors stated that the presence of microglia and choroid plexus cells in vhCOs provides a more complex microenvironment that contributes to neurogenesis, but this feature is not unique to the vascularized organoids, and they present no data showing that this contributes to neurogenesis. 

3) The authors suggest that the vhCOs had higher maturation scores in Fig. 2H-I, but this is not evident from Fig. 2I. Rather, the cells from the vhCO seem to be scattered around and may be more highly represented among the lower maturation scores. No statistical analysis was presented for these data. The authors also referenced Fig. S2E and claimed that the vhCOs are more mature, but this figure does not seem to compare vascular and non-vascularized hCOs.

4) The authors state that the vhCOs are electrophysiologically more mature because they were able to obtain Na+ currents and spontaneous action potentials at 90-100 days in vitro, earlier than in their prior work without endothelial cells. There was no parallel control with non-vascularized organoids, and therefore one cannot conclude that the presence of vasculature speeds functional maturation of neurons without direct batch-to-batch comparison.

5) The claim that this method is highly reproducible was not supported by the data. The authors note that they used 2 different human embryonic stem cell lines, but it is unclear which experiments were done with which cell line. Fig S1D shows both cell lines, but are all the other experiments from a single line? In addition, throughout the manuscript, details on how many times a certain experiment or differentiation were performed were not included. For example, what proportion of the vhCOs successfully exhibited vascular structures? To demonstrate reproducibility, multiple differentiations should be performed from multiple cell lines with consistent results.

6) For scRNAseq, how many organoids were analyzed? The experiment was not set up to examine variability between organoids or between batches of organoids as other published studies have done to look at reproducibility (e.g Velasco et al, Nature, 2019 and Yoon et al., Nature Methods, 2019). 

7) For electrophysiology, what was the success rate for patching cells? Of the cells that were successfully patched, what proportion had spontaneous activity and sodium currents?

8) There are a number of concerns with the in vivo grafting experiments. First, how many times were the grafting experiments done? In the few times that the authors state “n=…” it is unclear if the “n” refers to independent experiments or rather 3 samples from 1 experiment. For example, in Fig. 1M, n=3 samples may mean 3 organoids from a single differentiation, rather than 3 independent differentiations. For Fig 3E and 3G, n=6 was for 6 cells, which may be all from the same organoid. Second, the grafted organoid shown in Fig. 4 appears to be outside the parenchyma of the mouse brain, which it deforms from mass effect. Third, the authors claim laminar organization with CTIP2 being deeper and SATB2 being more superficial. This is not readily evident in Fig. 4B, and moreover, there appear to be DAPI+/CTIP2-/SATB2- rosette structures superficial to the demarcated SATB2 cells. If this recapitulated normal laminar organization, the rosette should be deep to the neurons. 

9) In the grafting experiment shown in Fig 4E, the authors state that there are GFAP+ cells in the grafts that appear to be derived from the human ES cells. They should discuss the discrepancy on why they find GFAP+ human cells in the grafts, but not in vitro when examined by scRNAseq. Furthermore, the GFAP staining in Fig 4E appears linearly aligned, which would be unusual for parenchymal astrocytes or astrogliosis, and instead may reflect radial glia - other markers for radial glia should be used to exclude this possibility. Also, the authors state that the myelinated fibers from the mouse are intruding into the graft. How can this be distinguished from the graft intruding into the mouse brain?

Other concerns:

10) The authors should discuss their choice of using HUVECs over other endothelial cells, including stem cell-derived endothelial cells, and in particular brain microvascular endothelial cells, which would seem to be a more appropriate choice.

11) In the introduction, the authors should consider citing cortical and cerebral organoid papers from the Sasai, Lancaster/Knoblich, Pasca, Song Labs (instead of or in addition to midbrain and hippocampal organoid papers). Also, the Mansour et al. paper from the Gage lab should be cite earlier in the introduction. The manuscript authors state that there has been no sophisticated vascular structure observed in brain-like organoids, but the Mansour et al. paper showed this. 

12) In Fig. 1C, the tube-like structures are not appreciated well, and in Fig. 1D, the TBR2+ layer appears thin. Also, it is unclear why there should be any cells triple-labeled with Tbr2/Sox2/IB4, as IB4 labels the vascular structures, not the intermediate progenitor cells.

13) In Figure 1E, Sox2 and p-VIM will label mitotic ventricular radial glia, and this is not specific to outer radial glia. The authors should use another marker such as HOPX for outer radial glia. Also, this staining was done at d65, an age at which on scRNA seq there were no HOPX+ outer radial glia. 

14) Fig 1K-M: Reelin and Cux1 immunolabeling would be helpful to show superficial cortical layer markers.

15) The lack of astrocytes on scRNA is puzzling. This should be confirmed with GFAP staining of the organoids.

16) In Fig. 2B, it appears that there are many more cells in the vOrganoids at d100 and d65 compared to the unvascularized organoids. This may be because the plot has opaque dots for the vOrganoids that are overlayed on top of the dots for the unvascularlized organoids. tSNE plots showing the vOrganoids and unvascularized organoids separately would help visualized these data. This also applies to Fig 2I. In addition, for the gene ontology analysis in Fig. 2F, are there several marker genes that fall under each category? If so, brackets should be used to show which rows of the heatmap correspond to the GO class. If not, having just 1 enriched synaptic gene (SYT1) does not make a compelling argument that vOrganoids are enriched in synaptic markers.

17) Fig 2D: The choice of comparing the organoid scRNAseq data to a dataset containing 2300 cells from human fetal prefrontal cortex from gestational ages (GW8-26) should be explained. The samples are restricted to a specific brain region. The dataset from Nowakowski et al., 2017 may be a better comparison.

18) Fig 2D and S2D: The interneuron populations of the organoids/vOrganoids do not overlap well with the human prefrontal cortex samples on the tSNE plot. The authors should comment on this and determine what genes may be differentially expressed between these two groups. Also, it is not clear if the interneurons derive from ganglionic eminence-like progenitors – markers for these (e.g., NKX2.1, DLX1/2, LHX6) should be examined by immunostaining and scRNAseq.

19) Fig 2H-I: The maturation trajectory does not appear to be tightly correlated compared to the cited study (Mayer et al, 2018).

20) For sodium/membrane current recordings (Fig 3A and S3D), additional steps to other voltages should be included (not just +20 and -20 mV).

21) The authors stated in the discussion “Pyramidal excitatory neurons and interneurons in the vOrganoids were aligned in layers that were similar to those of human neocortical lamination.” This was not demonstrated by the data.

22) Fig S1G: It is unclear what the white and black bars represent, and they are switched in position between the top and bottom panels.

23) In the asbstract, the “C” in sEPSCs and sIPSCs stands for current, not potential.

Reviewer #3: Shi et al develop a cerebral organoid differentiation protocol where they achieve generation of the vasculature (vOrganoids) by co-culturing HUVEC cells with human ESCs differentiated into cerebral organoids.

Overall, the manuscript is poorly prepared. Many descriptions lack proper quantifications, and many quantifications lack proper statistical reporting. There are numerous grammatical errors throughout the manuscript, and the logic of the experiments is difficult to follow.

There are several potentially interesting findings. For example, it is interesting that the vasculature localizes to the SVZ and observing spontaneous action potentials by day 90 suggests the vOrganoids mature well. Similarly, observing microglia in a forebrain protocol is unexpected and potentially useful.

However, my biggest concern that the authors have not really convincingly showed that their method of vOrganoids cultures have any major benefits over existing protocols.

Major concerns,

1) Despite a long introduction, the authors do a very poor job reviewing the state of the field of vascularization in organoids. For example, there is only a brief description of Mansour et al., 2018 in the discussion, and Pham et al., 2018 is only briefly mentioned in the introduction. Moreover Daviaud et al., 2018 is not really described. 

2) The main point of the article is the role of endothelial cells, yet there is very little characterization of these cells. These cells seem to have not been captured by the scRNA-seq, but could be characterized by other methods. In particular, HUVEC cells are likely very different from endothelial cells in the brain, but perhaps co-culture induces the HUVEC cells to a more brain like endothelial cell fate. This should be examined, and the authors should clearly acknowledge the differences between these cells and endothelial cells of the brain.

3) Throughout the manuscript, the authors provide insufficient information about many of the methods used for quantifying the phenotypes. Many descriptions appear to reflect some kind of quantification but no quantification or statistics is shown. For example, quantification of organoid growth presented in Figure S1 E – it is completely unclear from the figure description how many individuals were used, how many organoids, how were they selected, what statistical method was used for comparison, was there multiple hypothesis correction, what was the data distribution. This is just one of many examples where reporting on data quantification is unacceptable and makes it impossible for me to evaluate the statistical rigor in this manuscript.

4) In addition, I was expecting extensive characterization of the strengths and weakness of vOrganoids vs. conventional organoids, but the comparisons are limited. In particular, how much is cell death reduced in organoids, and do the vOrganoids survive transplant better than conventional organoids? For example, in In Figure S4 the authors show a Caspase3 stain but nothing to compare it too. Are there many Cas3 positive cells in organoids to begin with? Could they provide some quantification in organoids that were and were not transplanted into mouse? Similarly, is there more blood flow into vOrganoids than conventional organoids after transplant? Could the size increase of vOrganoids be driven by expansion of HUVECs themselves? There are some differentially expressed genes, but these should be validated by immuno and quantification across normal and vOrganoids.

5) At the end of the first results section the authors claim that “higher degree of vasculature assures sufficient oxygen and nutrient support for cell proliferation and differentiation.” However, it is unclear to me how this could possibly be achieved simply after co-culturing HUVECs. If the authors believe that this is the case they should address this experimentally. For example, if the hollow tubing of vasculature supplies more oxygen, this is testable.

6) Description of electrophysiological properties appears to be performed well, but it is unclear what novel observations are made. This component would be strengthened by experiments showing differences between the vOrganoids and standard organoids.

7) Most of the data showing human cerebral organoid cells after transplantation are very premature and mostly show the presence of different cell types that are normally found in an organoid after long enough culture. Integration into the mouse tissue is not novel and has been reported before.

Minor concerns

- Saying that organoids are laminated is quite a stretch. The reality is that there is a clear separation of progenitors vs neurons but within the neuronal layer having “cortical”-like layers has not been perfected. For the point of this article lamination is not relevant so I would recommend removing references to it. Moreover, their definition of upper layer neurons is SATB2+ cells, but this marker labels an identity (callosal neurons of all layers) not a layer. Similarly, CTIP2 labels subcerebral projection neurons.

- The appearance of microglia was reported in one other study (Ornel et al., 2018), but is a bit mysterious. Does this also occur in conventional organoids by this group or is it related to the HUVEC cells? Can the authors explain this result further? How are these distinguished from macrophages? Also, staining for AIF1 in figure 2 seems to be present almost everywhere in the field of view. It is unclear whether the staining was specific to AIF1.

- “The transplantation of cerebral organoids that are derived from hESCs or iPSCs into injured areas may be a promising therapy for improving neurologic deficits that are caused by trauma or neural degeneration”. I do not think there is a need for this sentence in the manuscript. What evidence do they have that transplanting an heterogeneous tissue (vs for instance transplanting pure neuronal subtypes) is preferable.

- Authors might consider a different nomenclature to refer to vascularized organoids. It is only natural to assume that abbreviations refer to regionalized organoids, so vOrganoids make it sound as “ventral” organoids, which is not what they intend. 

- The maturation analysis in figure S2 is confusing - choroid should be a separate lineage from excitatory neurons, and not included as a seeming progenitor cell

- In the methods, the co-clustering of scRNA-seq data is poorly explained. For example, what does a union of common highly variable genes mean? Is this the union or the intersection?

---

## [Decision Letter · Decision Letter 2]

31 Jan 2020

Dear Dr Wang,

Thank you very much for submitting a revised version of your manuscript "Vascularized human cortical organoids (vOrganoid) model cortical development in vivo" for consideration as a Methods and Resources at PLOS Biology. This revised version of your manuscript has been evaluated by the PLOS Biology editors, the Academic Editor and the original reviewers.

In light of the reviews (below), we are pleased to offer you the opportunity to address the remaining points from the reviewers in a revised version that we anticipate should not take you very long. Our Academic Editor advises that the following points are essential for the publication of the paper, whereas the other concerns not listed below are not required. 

1) The issue regarding caspase-3 antibody raised by reviewer 2;

2) Reviewer 2's suggestion on citing the new paper by Cakir et al (2019) and all this reviewer's minor concerns;

3) Reviewer 3's concern on Figure 2K and Table S1 - most part of this reviewer's concern #2;

4) Reviewer 3’s concern #3. Specifically, “clearly state what steps were taken to record from neurons of the same type in an unbiased way because differences could arise from recording from different neuron subtypes between conditions, or even neurons of different size. Is the result of a difference between vOrganoid and control organoids significant, or just a trend, no p-value is provided?“;

5) Part of reviewer 3's concern #4 : “a control experiment of whether induction occurs in HUVECs cultured on their own, and probably a control experiment that the P-glycoprotein antibody labels human HBMECs”;

6) All the minor concerns from reviewer 3. 

After re-submission, we will assess your revised manuscript and your response to the reviewers' comments and we may consult the reviewers again. We expect to receive your revised manuscript within 1 month.

Sincerely,

Di Jiang

PLOS Biology

REVIEWS:

Reviewer #1: Based on the authors' answers and new supporting information to our previous concerns for the manuscript, we feel that our concerns have been sufficiently addressed and would advise publication of this manuscript. This is paper will enrich the current literature on organoids and presents methodology for more physiological and meaningful study using this system.

Reviewer #2: This revised manuscript by Shi et al. employs human umbilical vein endotheial cells (HUVECs) co-cultured with human embryonic stem cells (hESCs) and human induced pluripotent stem cells (hIPSCs) to generate cerebral organoids. The authors have added substantial experimental data which bolster their findings and address most of the concerns brought up by the reviewers. They show that the vOrganoids have thicker neural progenitor zones, enhanced neurogenesis, and earlier maturation of electrophysiological properties. They also provide evidence supporting the reproducibility of their data by using 2 hESC and 2 hIPSC lines to generate vOrganoids, and they give additional information about the number of replicates and their statistical analyses.

There are several remaining concerns:

1) The authors demonstrate decreased caspase-3 staining in their vOrganoids compared to controls to support less cell death, but they use an antibody recognizing full length, rather than cleaved, caspase-3. Full length caspase-3 may not accurately reflect apoptotic cell death and measurement of cleaved caspase-3 should be used instead. 

2) While the single cell RNA-seq data (Figure 2) demonstrate cell type diversity in the vOrganoids, these data do not make a compelling case that the vOrganoids are an improvement over non-vascularized ones. The main differences are that the vOrganoids have increased NEUROD2+ neurons, which may suggest that neurogenesis is promoted. An increase in NEUROD2+ neurons is not necessarily advantageous. The authors also note that there are differentially expressed genes and that on GO analysis there are some enriched pathways, but again this does not show any advantage (e.g. better recapitulation of developing human fetal brain, or improved cell type diversity) with using HUVECs.

3) In the time that the authors revised this manuscript, an additional publication using ETV2-expressing hESCS to generate cerebral organoids with a vascular system was published (Cakir et al, Nature Methods, vol 16, November 2019, 1169-1175). The authors should reference this paper in their introduction and discuss/compare this paper with the present results in the conclusion of this manuscript.

Additional minor concerns:

1) Several abbreviations (e.g. vRG, oRG) remain undefined in the text.

2) The authors responded that they toned down the claims of lamination in the organoids, but in the abstract, they still state that their model recapitulates lamination of the neocortex. This statement should be removed.

3) In lines 100-101, the authors state that "all of these studies indicate that vascularization is important for organoid survival." This is overstated, as there are plenty of protocols that do not have vascularization but have good organoid viability.

4) For Fig S1G, it would be preferable if the data were presented as individual data points with mean/standard deviation rather than a bar graph, to better demonstrate the variability/range of the data.

5) The claim that vOrganoids are larger than controls (Fig. S1H) is questionable. The error bars appear very small compared to what might be expected for typical size variability of organoids--are the error bars standard deviation or standard error of the mean? Which statistical test was performed? It would be appropriate to do 2-way ANOVA with multiple comparisons, rather than do repeated t-tests. Even if the data are statistically significant, the relative difference in size does not appear to be of biological significance.

6) For Fig. S1L, there is no control staining—perhaps the HUVECs express p-gp even in the absence of being cultured with the organoid? 

7) The calcium imaging data in Fig 3I was normalized to 1 for both the d50 and d85 conditions. In order to see relative differences between d50 and d85 (one would expect increased activity with older organoids), the data should be normalized to just the day 50 organoid in ACSF.

8) The authors should provide additional explanation regarding the response of calcium transients to pharmacological treatments of the organoids (Fig. 3I and lines 274-281). Why are CNQX and APV more effective at reducing calcium transients at the later time point? The authors should also explain in more detail the effect of BMI. The effect at d50 of decreasing calcium transients may be a developmental effect, where opening of GABA-A receptors can depolarize the cell membrane, since the chloride ion concentration is higher inside the cell compared to outside the cell. The effect at d85 (where there does not seem to be an effect of the BMI), may be the transition point when the intracellular chloride ion concentration is decreasing, and opening of the GABA-A receptors may not have an effect on the membrane potential.

9) Figure S3H and I are discordant with each other. Why do the tracings in the ACSF condition in panel H show calcium transients, but the ACSF in panel I has even less activity than the amount seen in the TTX condition of panel H?

10) For Figure 4C, one cannot conclude that there is synaptic connectivity between the organoid graft and the host brain. The co-localization of pre-synaptic and post-synaptic markers could also be mouse-to-mouse or human-to-human.

Reviewer #3: Manuscript by Shi et al., presents a vascularized organoid model that could potentially serve as an improved system for modeling aspects of cortical brain development, for transplanting organoids into mice, and for treating brain disorders. The revision attempts to more explicitly address whether and in what ways the vOrganoids are an improvement over existing organoid models. The manuscript is improved in this respect. For example, I find the improved grafting of vOrganoids compared with control organoids and the mixing of HUVEC derived and mouse vascular cells particularly interesting. However, I still have major concerns that I address below, in the context of each stated improvement.

1) The study describes "improved cell survival in vOrganoids during culture (new Fig S1F-G) and larger size (new Fig.S1H)"

Figures S1F and S1G are key to showing improvements in vOrganoids and should probably be in the main text. However, understanding the mechanism of these improvements would be particularly powerful, especially since the authors discuss possibilities of nutrients, signaling and oxygen several times. Therefore, it would be really add to the manuscript's impact if the authors could test whether the vasculature increases oxygen tension inside the organoids (thereby influence the HIF1a protein levels, as performed in Pasca 2019.

Also, can the authors rule out that the increased size is not simply driven by the number of HUVECs transplanted and their proliferative potential?

2) The study states scRNA-seq suggests improvement in gene expression and neurogenesis in vOrganoids

I'm particularly concerned about the scRNA-seq data analysis. The interpretation of clustering is useful and very interesting that subcortical cell types emerge. However, these datasets could allow for a quantification of whether vOrganoid cells of particular types more closely resemble fetal cell counterparts than control organoids.

Instead, the study simply implies the vOrganoid cells are improved, but only shows that you can integrate the organoid cells with primary cells using a batch correction algorithm. However, these batch correction algorithms can overcome severe differences between conditions, and it is unclear whether the integration is superior with vOrganoid cells.

Figure 2K could further help address if vOrganoids bring cells closer to fetal cell gene expression if the differential expression between vOrganoids were put in context of fetal cell data.

However, I have major reservations with this analysis and with Table S1. First, it is unclear what cell types are listed in the Fig 2K table. Is there a cell cycle gene expression difference between progenitors across vOrganoid and control conditions, or are there more progenitor cells in the control organoids? A meaningful comparison must analyze cells of the same type, but the heatmap looks like some sort of composition difference between the control and vOrganoid cells analyzed, with more dividing progenitors in the control organoids.

In this regard, the enrichments in Figure 2K and Table S1 are incomplete, unconvincing, and potentially misleading. The enrichments of vOrganoid cells in "axon guidance," "blood vessel morphogenesis" and "cellular response to oxygen levels" sound at first like they represent a more fetal-like state (and a quick read of the manuscript would interpret this analysis as evidence of improvements in vOrganoids, but the genes listed in the table for enrichments are either very few (3 genes in total account for multiple statements about cellular response to oxygen!) or indicative of an artifact (6/7 genes listed for axon guidance are ribosomal protein transcripts that often reflect outlier cells to be removed in scRNA-seq QC!). The p-values for the GO terms also must be corrected for multiple hypothesis testing.

This analysis should be dropped or completely re-done, explicitly addressing how cells of the same type differ across conditions and whether these differences produce more fetal-like cells in vOrganoids.

3) The study describes earlier maturation of the active electrophysiological activities (new Fig 3A-B and Fig.S3A-B); 

The study should clearly state what steps were taken to record from neurons of the same type in an unbiased way because differences could arise from recording from different neuron subtypes between conditions, or even neurons of different size. Is the result of a difference between vOrganoid and control organoids significant, or just a trend, no p-value is provided? The other physiological metrics are interesting background, but it must be made very clear that these metrics were not compared to control organoids and that it is therefore unclear whether vascularization influenced these features

4) The study suggests the HUVEC cells change to resemble human brain vasculature, potentially indicating an unappreciated utility for this cell type which also has the advantage of being easy to access

However, the observed induction of P-glycoprotein experiment in HUVECs needs a control experiment of whether induction occurs in HUVECs cultured on their own, and probably a control experiment that the P-glycoprotein antibody lables human HBMECs. The interpretation would also be more believable if a range of HBMEC markers were induced instead of just one.

Minor comments:

-abstract still says "recapitulate lamination of neocortex" but there is no evidence for cortical layers resembling in vivo lamination

-The data in figure panels does not support the text that "later-born superficial layer neurons (SATB2+cells) were more superficially localized to the CTIP2+ neurons" - certainly the SATB2 neurons accumulate superficial to SOX2 and TBR2 but evidence for cortical plate lamination is weak in organoid field beyond some coarse SATB2 CTIP2 stains that lack quantification across replicates, individuals, and developmental time. Even the TBR1 staining in S1O extends to the surface and there is RELN staining very interior. Regardless, without comparing quantitative cortical plate lamination phenotypes in vOrganoids to control organoids, it is impossible to know whether the resolution of layers improves by this treatment.

In addition, while it is true that virtually all upper layer projection neurons are SATB2+, over 30% of deep layer neurons project through the corpus callosum and are SATB2+ (refer to McConell, Chen, Macklis, and Arlotta labs literature). I would encourage the author to either refer to these cells as callosal neurons or use markers with a more clear laminar restriction such as CUX1 and CUX2 (for example Quadrato et al.)

-Figure S1 E show less extensive IB4 staining in iPSC-derived organoids and Figure S1F shows much of the IB4 localization on the surface of older organoids (D115), which differs from the internal staining in Figure 1G (d45, d65). From the discussion, it sounds like the vasculature being precisely in the VZ/SVZ during early culture stages - can the authors clarify how the HUVEC distribution changes over time and how consistent the transition is?

-In Fig 2 - why use both UMAP and PAGA for visualization of dimensionality reduction? Also, explain what PAGA stands for and how it is performed. In Figure 2, explain the method for batch integration and how it was corroborated by other methods. 

-What is the physiological significance of increased outward current in vOrganoids and what other properties might be expected to change along with this one?

-it is uncertain whether there is synaptic connectivity between the mouse brain and organoid cells based on these studies, although other reports (Mansour, Vanderhagen, de Paola indicate this occurs)

-It is very interesting that "the vascular systems in the vOrganoids could improve the cell survival in the host" - Fig 4E is exciting, providing evidence for the host vasculature connecting to the HUVECs, which could explain the improvements in transplant of vOrganoids

-This study observes a 40% Caspase3/DAPI staining in grafted organoids that lack HUVEC cells, which seems very high compared to Mansour et al 2018 who observe almost no TUNEL staining in day 50 and day 233 post-impantation organoids - is there an interpretation for this different observation? The Mansour paper makes it sound like grafting a normal organoid already solves the apoptosis problem, but that does not seem to be the case here. Because the improved grafting of vOrganoids may be a key feature, it would be important to address this discrepancy in some way, even just in the text.

-Figure 4B organoid staining does not resemble cortical plate lamination or cortical folding. There are more clusters of SATB2 by the surface, but these do not form a layer, and as acknowledge, these are interrupted by VZ-like rosettes, which would account for the bumpiness described as cortical folding. Strong claims of cortical layer formation and cortical folding require strong evidence, including clear definitions and quantification which are lacking, and it is unhelpful for the field to promote confusion here. Also, it is unclear how these observations compare to control organoids.

---

## [Editor Report · Decision Letter 3]

10 Mar 2020

Dear Dr Wang,

Thank you for submitting your revised Methods and Resources entitled "Vascularized human cortical organoids (vOrganoid) model cortical development in vivo" for publication in PLOS Biology. I have now obtained advice from the Academic Editor who has assessed your revision. 

We're delighted to let you know that we're now editorially satisfied with your manuscript. However before we can formally accept your paper and consider it "in press", we also need to ensure that your article conforms to our guidelines. A member of our team will be in touch shortly with a set of requests. As we can't proceed until these requirements are met, your swift response will help prevent delays to publication. Please also make sure to address the data and other policy-related requests noted at the end of this email.

*Copyediting*

*Published Peer Review History*

*Early Version*

*Submitting Your Revision*

Sincerely,

Di Jiang

PLOS Biology

ETHICS STATEMENT:

-- Please consolidate your ethics statements currently in "Cell lines" and "Tissue sample collection" subsections into a separate subsection entitled "Ethics Statement" and place it in the beginning of the Methods section.

-- Please include the full name of the IACUC/ethics committee that reviewed and approved the animal care and use protocol/permit/project license. Please also include an approval number.

-- Please include the specific national or international regulations/guidelines to which your animal care and use protocol adhered. Please note that institutional or accreditation organization guidelines (such as AAALAC) do not meet this requirement.

-- Please include information about the form of consent (written/oral) given for research involving human participants. All research involving human participants must have been approved by the authors' Institutional Review Board (IRB) or an equivalent committee, and all clinical investigation must have been conducted according to the principles expressed in the Declaration of Helsinki.

DATA POLICY:

-- Regardless of the method selected, please ensure that you provide the individual numerical values that underlie the summary data displayed in the following figure panels as they are essential for readers to assess your analysis and to reproduce it: Figures 1M, 2ABCI, 3ABDFI, S1GHJK, S2ABCDEGHIJMN, S3ABEFG, S4B. NOTE: the numerical data provided should include all replicates AND the way in which the plotted mean and errors were derived (it should not present only the mean/average values).

-- Please provide an editor/reviewer key/token to your RNA sequencing data in GEO, GSE131094, which will allow us to check the data before accepting the paper.

---

## [Editor Report · Decision Letter 4]

17 Apr 2020

Dear Dr Wang,

On behalf of my colleagues and the Academic Editor, Bing Ye, I am pleased to inform you that we will be delighted to publish your Methods and Resources in PLOS Biology. 

Early Version

PRESS 

Kind regards,

Vita Usova

Publication Assistant, 

PLOS Biology

on behalf of

Di Jiang,

Associate Editor

PLOS Biology